# Multiphoton imaging of neural structure and activity in *Drosophila* through the intact cuticle

**Max Jameson Aragon[1†‡], Aaron T Mok[2†], Jamien Shea[1†], Mengran Wang[2†], Haein Kim[1], Nathan Barkdull[3], Chris Xu[2], Nilay Yapici[1]***

[1]Department of Neurobiology and Behavior, Cornell University, Ithaca, United States; [2]School of Applied and Engineering Physics, Cornell University, Ithaca, United States; [3]Department of Physics, University of Florida, Gainesville, United States

**ABSTRACT** We developed a multiphoton imaging method to capture neural structure and activity in behaving flies through the intact cuticle. Our measurements showed that the fly head cuticle has surprisingly high transmission at wavelengths >900nm, and the difficulty of through-cuticle imaging is due to the air sacs and/or fat tissue underneath the head cuticle. By compressing or removing the air sacs, we performed multiphoton imaging of the fly brain through the intact cuticle. Our anatomical and functional imaging results show that 2- and 3-photon imaging are comparable in superficial regions such as the mushroom body, but 3-photon imaging is superior in deeper regions such as the central complex and beyond. We further demonstrated 2-photon through-cuticle functional imaging of odor-evoked calcium responses from the mushroom body γ-lobes in behaving flies short term and long term. The through-cuticle imaging method developed here extends the time limits of in vivo imaging in flies and opens new ways to capture neural structure and activity from the fly brain.

**\*For correspondence:**
ny96@cornell.edu

†These authors contributed equally to this work

**Present address:** ‡Princeton Neuroscience Institute, Princeton University, Princeton, United States

**Competing interest:** The authors declare that no competing interests exist.

## Editor's evaluation

This study reports a way to record neural structures and activity in behaving *Drosophila* for up to 12 hours, without removing the cuticle. This opens the way to longer and more intact recordings, including post-imaging recovery of the flies. The authors describe their method for compression of air sacs to achieve this, and characterize image quality and responses using this method.

## Introduction

Animal nervous systems across lineages have evolved to solve many of the same problems; foraging for food and water, finding mates to reproduce, and avoiding predators to stay alive. They navigate their environment via coordinated movements and learn and remember the relative values of sensory stimuli around them to maximize their fitness and survival. At each instant in time, an animal must evaluate external sensory information and its current behavioral state to decide what to do next (***Dickson, 2008***; ***Hunt and Hayden, 2017***; ***Lütcke et al., 2013***; ***Tinbergen, 1969***). A major technological challenge to revealing how the brain encodes behavioral states in real-time is that even the simplest neural computation involves interactions across the nervous system at various time scales, while our tools for assessing neural activity are restricted in time and space because of the currently available imaging sensors, methods, and preparations (***Lerner et al., 2016***). Optical methods remain the most established and fruitful path for revealing population dynamics in neural circuits at long time scales

(ranging from minutes to hours) by providing high temporal and spatial resolution measurements (*Ji et al., 2016*; *Luo et al., 2018*; *Svoboda and Yasuda, 2006*).

The fly, *Drosophila melanogaster*, offers an ideal experimental system to investigate neural correlates of behavioral states and decisions because of its compact nervous system and diverse state-dependent behaviors that it executes in response to sensory stimuli (*Barron et al., 2015*; *Dickson, 2008*). To understand how molecularly defined neural circuits evaluate sensory information in different behavioral states, it is critical to capture the activity of populations of neurons over long time scales as flies are changing their physiological needs (*Luo et al., 2018*; *Simpson and Looger, 2018*). These functional imaging experiments require imaging preparations, which should allow chronic neural activity imaging for at least 12 hr. Current methods used in fly optical physiology require the fly head cuticle, trachea, and fat body to be removed by microsurgery to provide optical access to the nervous system (*Grover et al., 2016*; *Minocci et al., 2013*; *Seelig et al., 2010*; *Sinha et al., 2013*; *Wang et al., 2003*). These preparations are limited in imaging duration because, after some time, the brain tissue starts to degenerate due to damaged circulation resulting from the cuticle removal surgery. For example, with current imaging preparations, fly olfactory neurons show reliable $Ca^{2+}$ responses for four to five hours after surgery (*Wang et al., 2003*). An imaging method in which the head cuticle is intact, thereby eliminating the need for traumatic head surgery before functional imaging, is essential for advancing fly neuroscience research in the direction of chronic recordings of neural activity during ongoing behaviors. This includes being able to image the same fly brain across multiple days. In mice, multiday imaging experiments are achieved by implanting a cranial window following removal of part of the skull (*Hefendehl et al., 2012*; *Trachtenberg et al., 2002*). Similar imaging preparations have been developed for flies (*Grover et al., 2016*; *Huang et al., 2018*; *Sinha et al., 2013*). However, because imaging window implantation requires a tedious surgery with low success rates and complications that occur afterwards, these methods are not commonly used. A recent development in multiphoton imaging is the use of long-wavelength lasers in 3-photon (3 P) microscopy which improves signal-to-background ratio (SBR) by several orders of magnitude compared to current 2-photon (2 P) imaging methods (*Horton et al., 2013*; *Wang et al., 2018a*; *Wang et al., 2018b*). While 3 P microscopy with 1700 nm excitation of red fluorophores and adaptive optics has shown promising results in imaging the fly brain through the cuticle (*Tao et al., 2017*), it is not clear if the technique is widely applicable to common blue and green fluorophores with much shorter excitation wavelengths (e.g. 1320 nm).

Here, we developed a method for imaging fly neural structure and activity through the intact head cuticle using both 2P and 3P microscopy. We first measured the ballistic and total optical transmission through the dorsal fly head cuticle and surprisingly found that the head cuticle has high transmission at the wavelengths that are used to excite green fluorophores in 2P and 3P microscopy (~920 nm and ~1320 nm, respectively). We showed that the tissue that interferes with the laser light and limits imaging through the cuticle into the brain is not the head cuticle but the air sacs and the tissue underneath the cuticle. Next, we developed fly preparations by either compressing the air sacs or removing them from the imaging window, allowing through-cuticle imaging of the fly brain. Using these imaging preparations, we performed deep, high spatial resolution imaging of the fly brain and determined the attenuation length for imaging through the cuticle with 2P (920 nm) and 3P (1320 nm) excitation and compared our results to cuticle-removed preparations. Our measurements showed that 2P and 3P excitation performed similarly in shallow regions (i.e. in the mushroom body) of the fly brain, but 3P excitation at 1320 nm was superior for imaging neural activity and anatomical features in deeper brain structures (i.e. in the central complex). Furthermore, using 2P and 3P excitation, we recorded food odor-evoked neural responses from the Kenyon cells comprising the mushroom body γ-lobes using a genetically encoded $Ca^{2+}$ indicator, GCaMP6s (*Chen et al., 2013*). In our simultaneous 2P and 3P functional imaging experiments, we found no differences between 2P and 3P excitation, while recording odor-evoked responses from the mushroom body γ-lobes through the cuticle. To demonstrate that our cuticle-intact imaging method can be used for recording neural activity in behaving flies, we used 2P excitation and captured odor-evoked neural responses from mushroom body γ-lobes in flies walking on an air-suspended spherical treadmill. Finally, we demonstrated long-term functional imaging by reliably capturing odor-evoked neural responses from γ-lobes with 2P excitation for 12 consecutive hours. The cuticle-intact imaging method developed here allows multiphoton imaging of the fly brain through the head cuticle opening new ways to

capture neural structure and activity from the fly brain at long time scales and potentially through the entire lifespan of flies.

## Results

### Fly head cuticle transmits long-wavelength light with high efficiency

To develop a cuticle-intact imaging method using multiphoton microscopy, we first measured light transmission at different wavelengths through the fly head cuticle. Previous experiments showed that, within the wavelength range of 350–1000 nm, the relative transmission of the dorsal head cuticle of *D. melanogaster* improves with increasing wavelengths (*Lin et al., 2015*). However, the absolute transmission, which is critical for assessing the practicality of through-cuticle imaging, was not reported. In our experiments, we quantified both the total and ballistic transmission of infrared (IR) laser lights through the cuticle using the setup from our previous work (*Mok et al., 2021*). Dissected head cuticle samples were mounted between two glass coverslips and placed in the beam path between the laser source and the photodetector (*Figure 1A*). The total and ballistic transmission through the cuticle samples were measured using a custom-built system (*Figure 1B*). For ballistic transmission, light from a single-mode fiber was magnified and focused on the cuticle with a ~25 μm spot size. *Figure 1C* illustrates the light path of ballistic transmission experiments. The sample stage was translated to obtain measurements at different locations on the head cuticle. Ballistic transmission through the cuticle was measured at seven different wavelengths (852 nm, 911 nm, 980 nm, 1056 nm, 1300 nm, 1552 nm, 1624 nm) that match the excitation wavelengths for typical 2 P and 3 P imaging. We found that for all the IR wavelengths tested, the ballistic transmission through the cuticle was high, reaching >90% at 1300 nm (*Figure 1D*, *Figure 1—source data 1*). Since fluorescence signal within the focal volume in 2P and 3P microscopy is mostly generated by the ballistic photons (*Dong et al., 2003*; *Horton et al., 2013*), our results showed that ballistic photon attenuation by the fly cuticle does not limit multiphoton imaging through the intact cuticle.

To assess the absorption properties, we measured the total transmission through the head cuticle. For these measurements, laser light from a single mode fiber was magnified and focused on the cuticle sample with a ~50 μm spot size (*Mok et al., 2021*). An integrating sphere (IS) was placed immediately after the cuticle to measure the total transmission. *Figure 1F* illustrates the light path of total transmission experiments. Total transmission through the cuticle was measured at nine different wavelengths (514 nm, 630 nm, 852 nm, 911 nm, 980 nm, 1056 nm, 1300 nm, 1552 nm, 1624 nm). The shorter wavelengths of 514 nm and 630 nm were chosen to match the typical fluorescence emission wavelengths of green and red fluorophores. Similar to the ballistic transmission experiments, we found that the total transmission generally increased with wavelength (*Figure 1G*, *Figure 1—source data 1*), and the total transmission for both the green and red wavelengths was sufficiently high ( >60%) for practical epi-fluorescence imaging using 2 P or 3 P excitation. We also scanned the cuticle with a motorized stage in the setup at selected wavelengths (*Figure 1E and H*), and these spatially resolved transmission maps confirmed that there are only a few localized regions at the periphery of the cuticle with low transmission. Our results demonstrated that absorption and scattering of long-wavelength light by the *Drosophila* head cuticle is small, and cuticle-intact in vivo imaging of green (e.g. green fluorescent protein (GFP) and GCaMP) and red fluorophores (e.g. red fluorescent protein (RFP), and RCaMP) through the intact cuticle is possible in adult flies using 2 P or 3 P excitation.

### Through-cuticle multiphoton imaging of the fly brain

Based on our cuticle transmission results, we developed a cuticle-intact imaging method where we either used head compression to minimize the volume of the air sacs (*Figure 2A*, *Video 1*) or removed them completely from the head capsule (*Figure 2—figure supplement 1B-D*). Using our new fly preparations, we imaged the fly brain through the cuticle with no head compression, semi compression, or full compression (*Figure 2B–D*). We expressed membrane-targeted GFP (CD8-GFP) selectively in mushroom body Kenyon cells and scanned the fly brain through the cuticle using 2 P and 3 P excitation at 920 nm and 1320 nm, respectively (*Figure 2E–G*). Kenyon cells are the primary intrinsic neurons in the insect mushroom body. Diverse subtypes of Kenyon cells (n = ~2200) extend their axons along the pedunculus and in the dorsal and medial lobes (*Crittenden et al., 1998*; *Ito et al., 1998*; *Strausfeld et al., 1998*). These neurons receive and integrate information from heterogeneous

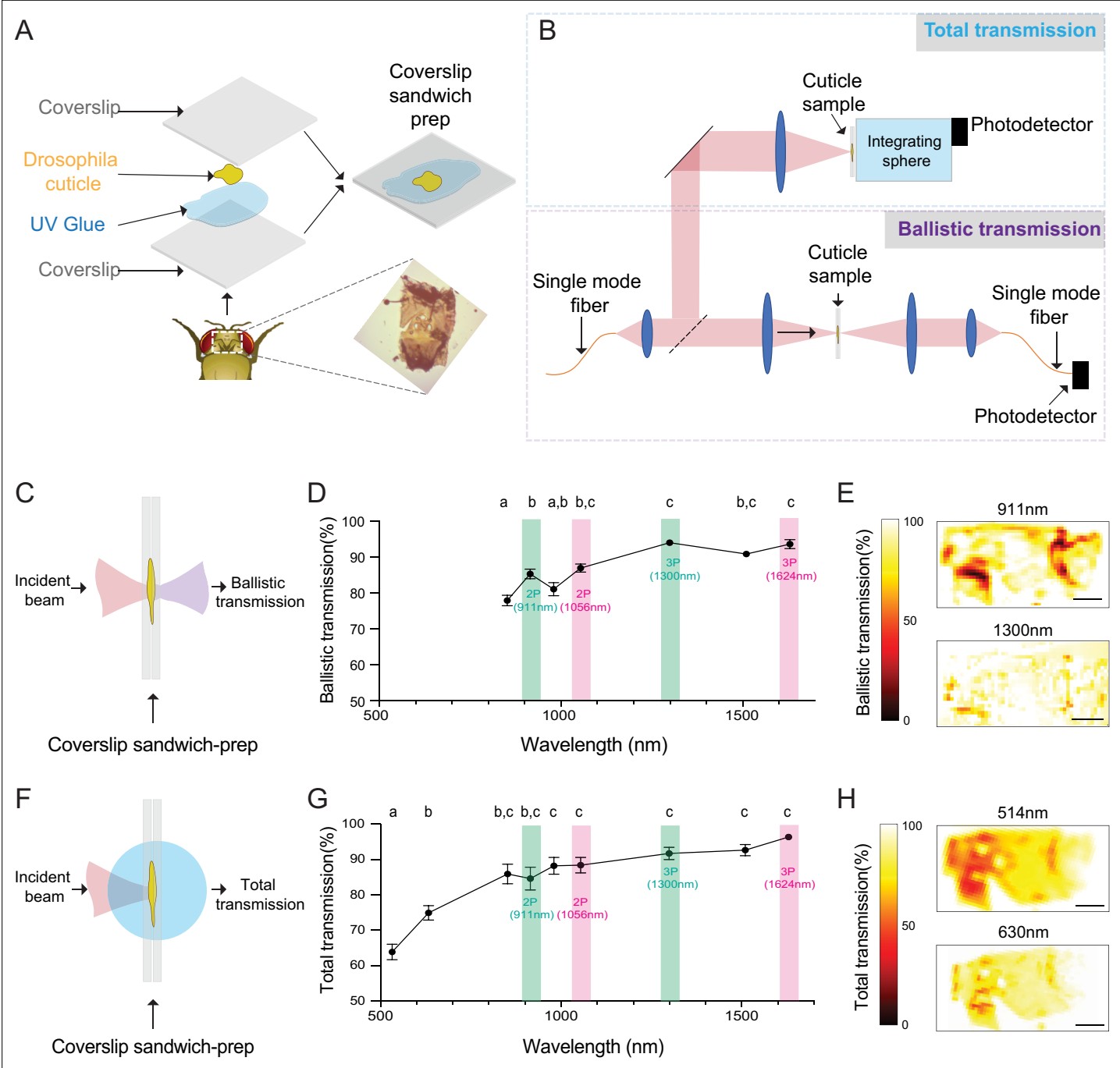

**Figure 1.** Ballistic and total optical transmission of the fly head cuticle. (**A**) Schematic of the cuticle preparation. (**B**) Schematic of the cuticle optical transmission measurement setup. (**C**) Schematic of the ballistic optical transmission through the cuticle. (**D**) Results of the ballistic optical transmission experiments at various wavelengths (n = 56 measurements at each wavelength, 5 different samples). (**E**) Spatially resolved maps at 911 nm and 1300 nm with the percent ballistic transmission color-coded. Lighter colors indicate higher transmission and darker colors indicate lower transmission. (**F**) Schematic of the total optical transmission through the cuticle. (**G**) Results of the total optical transmission experiments at various wavelengths (n = 20 measurements at each wavelength, 4 different samples). (**H**) Spatially resolved maps at 514 nm and 630 nm with the percent total transmission color-coded. Lighter colors indicate higher transmission and darker colors indicate lower transmission. One-way ANOVA with post hoc Tukey's test. Data points labeled with different letters in D and G are significantly different from each other (scale bars = 100 µm).

The online version of this article includes the following source data for figure 1:

**Source data 1.** Source data for plots *Figure 1D and G*.

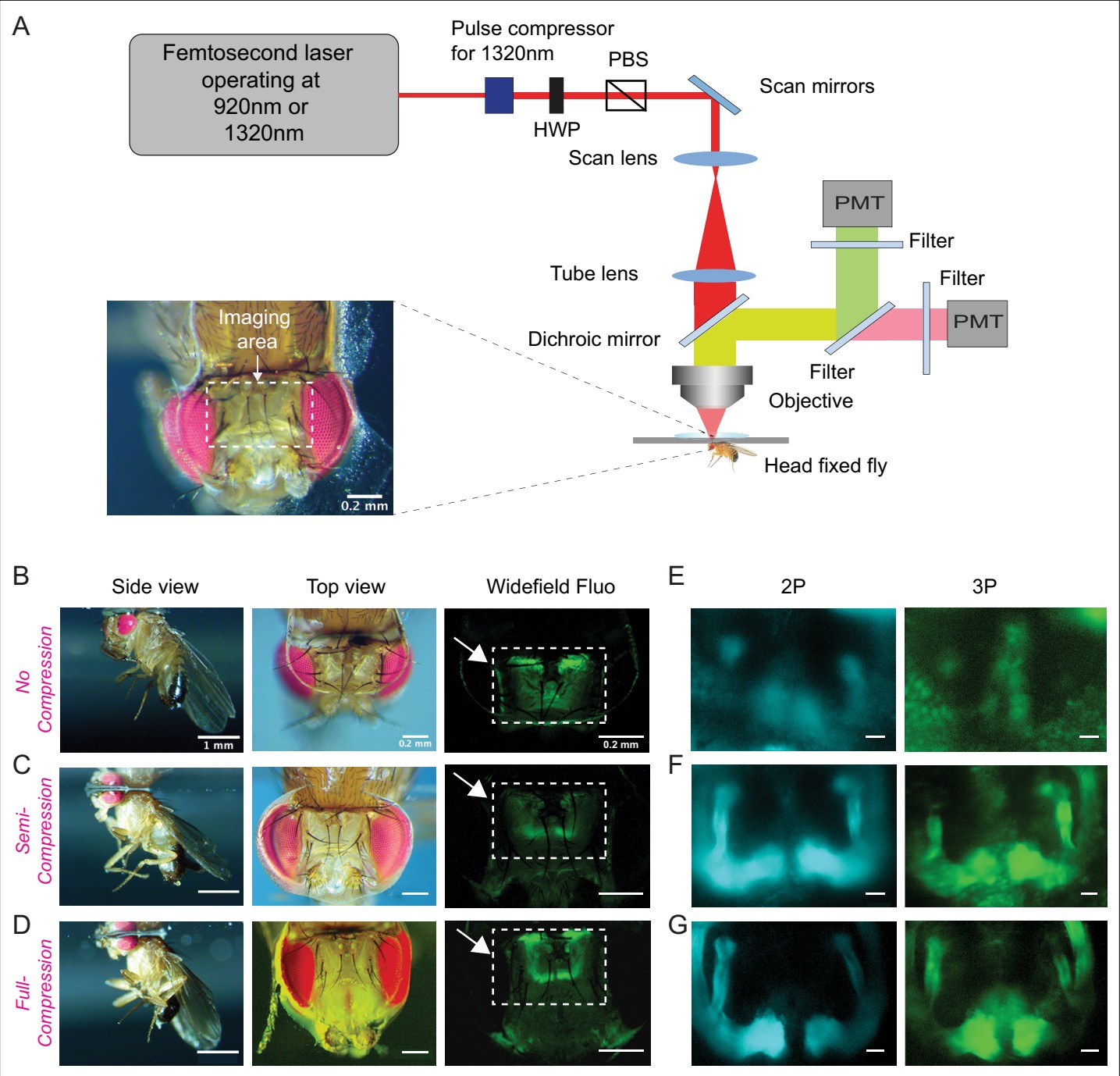

**Figure 2.** Through-cuticle imaging of the fly brain with 2P and 3P excitation. (**A**) Schematic of the multiphoton microscope setup. Fly head is fixed to a cover slip and placed under the objective (HWP, half-wave plate; PBS, polarization beam splitter; PMT, photomultiplier tube). The imaging window on the fly head is shown in the picture (lower left). Scale bar = 200 μm. (**B–D**) The head-uncompressed and head-compressed imaging preparations. The first column shows the side image of the fly that is head fixed to the cover glass (scale bar = 1 mm). The second and third columns show the fly head visualized under a brightfield (top view) and fluorescent dissecting microscopes (widefield-fluo), respectively. Arrows and the rectangle area in widefield-fluo column indicate the imaging window (scale bar = 200 μm). (**E–G**) Cross-section imaging of the mushroom body Kenyon cells expressing CD8-GFP through the head cuticle at 920 nm (2P) and 1320 nm (3P) excitation. The Z projections of 2P (cyan, left) and 3P (green, right) imaging stacks. For each imaging preparation, the same fly head is imaged with 3P and 2P excitation (scale bar = 20 μm).

The online version of this article includes the following figure supplement(s) for figure 2:

**Figure supplement 1.** Head compression does not affect male courtship and removal of air sacs allows 2P and 3P imaging in a head-uncompressed preparation.

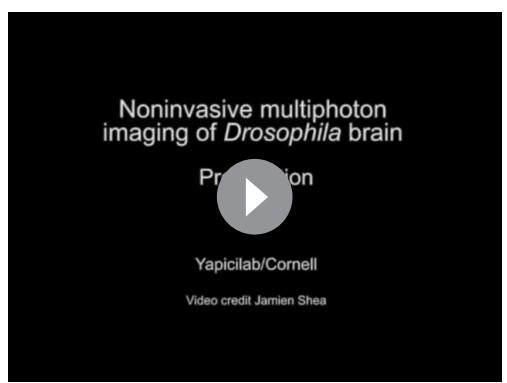

**Video 1.** Video demonstrating how to prepare flies for through-cuticle imaging.

https://elifesciences.org/articles/69094/figures#video1

sets of projection neurons which carry olfactory, gustatory, and visual sensory information (*Owald and Waddell, 2015*; *Yagi et al., 2016*). Kenyon cell dendrites arborize in the calyx, while their axons fasciculate into anatomically distinct structures called lobes, with the dorsal lobes forming α and α′ branches, and the medial lobes containing β, β′, and γ branches (*Crittenden et al., 1998*; *Ito et al., 1998*; *Zheng et al., 2018*). We used transgenic flies that specifically expressed membrane-targeted GFP in Kenyon cells forming α, β, and γ lobes (*Krashes et al., 2007*). In noncompressed flies, the mushroom body lobes were barely visible in both 2 P and 3 P imaged flies. Compressing the head against the cover glass with forceps during the curing process drastically improved image quality (*Figure 2E–G*), mushroom body lobes were visible in the semi compressed and full compressed preparations in both 2 P and 3 P imaged flies. Based on our observations of the leg movements, flies behaved similarly with no head compression or in semi compressed preparations but not in full compression. We also tested the male courtship behavior of flies whose heads were previously semi compressed. Our results showed that semi head compression does not affect male courtship behavior grossly; head-compressed males are able to copulate with females at similar rates as control males (*Figure 2—figure supplement 1A*). Based on our imaging and behavior results, we decided to use the semi compressed preparation in our experiments.

Why does head compression improve image quality during 2 P and 3 P imaging? We hypothesized, head compression might reduce the volume of air sacs and the surrounding tissue between the cuticle and the brain, allowing better transmission of long wavelength laser light through these structures. To test our hypothesis, we surgically removed air sacs from one side of the fly head and imaged the brain using 2 P and 3 P excitation without any head compression. As predicted, we were able to image the mushroom body lobes on the side where air sacs were removed but not on the side where intact air sacs were present (*Figure 2—figure supplement 1B-D*, *Video 2*). Our results demonstrated that the tissue that interferes with 2 P and 3 P laser light is not the cuticle itself but the air sacs and other tissues that are between the head cuticle and the brain.

## Comparison of 2P and 3P excitation for deep brain imaging through the fly head cuticle

Our experiments showed that through-cuticle imaging is possible with both 2 P and 3 P excitation. In general, 3 P excitation requires higher pulse energy at the focal plane compared to 2 P excitation

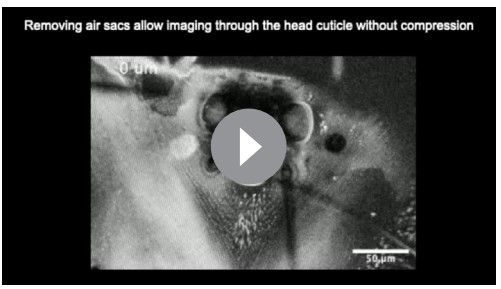

**Video 2.** Z stack of the mushroom body Kenyon cells expressing GCaMP6s. Imaging is done through the head cuticle using 1320 nm (3P) excitation after removing air sacs only on one side of the head (scale bar = 50 µm, no head compression).

https://elifesciences.org/articles/69094/figures#video2

because of the higher-order nonlinearity. On the other hand, longer wavelength (1320 nm) used for 3 P excitation can experience less attenuation while traveling in the brain tissue leading to increased tissue penetrance and imaging depth (*Wang et al., 2018a*). To compare the performance of 2 P and 3 P excitation for through-cuticle imaging, we imaged the entire brain in a fly expressing membrane-targeted GFP pan neuronally. *Figure 3A* shows the images from the same fly brain at different depths obtained with 2 P (920 nm) and 3 P (1320 nm) excitation. At the superficial brain areas such as the mushroom bodies, 2 P and 3 P excitation performed similarly. As we imaged deeper in the brain, 3 P excitation generated images with higher contrast compared

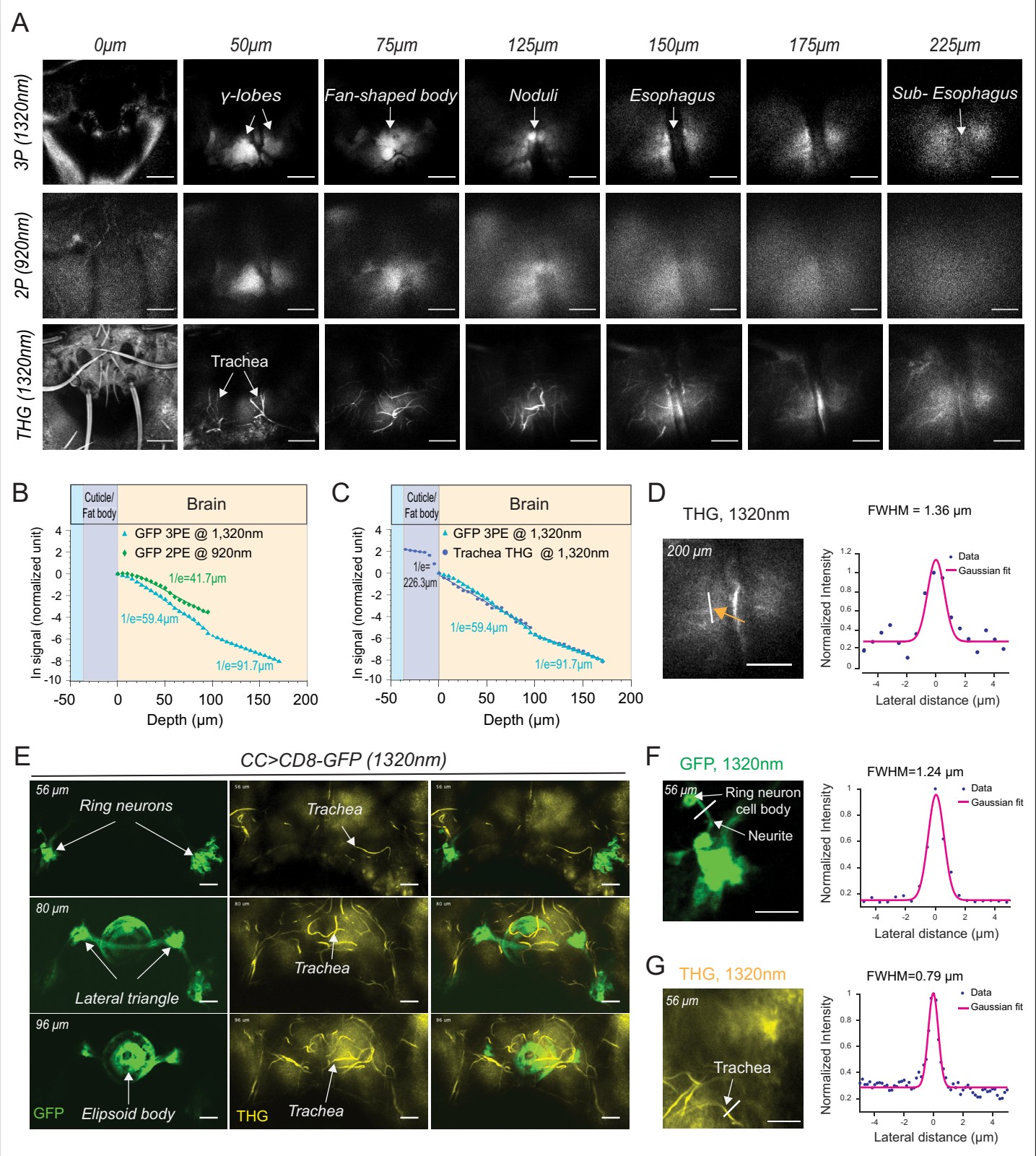

**Figure 3.** 2P and 3P structural imaging of the fly brain. (**A**) Cross-section images of the fly brain through the cuticle with 3P (top) and 2P (bottom) excitation at different depths. The third harmonic generation (THG) images are included at the bottom. 3P excitation power is <11 mW and the repetition rate is 333 kHz. 2P excitation power is <15 mW and the repetition rate is 80 MHz, scale bars = 50 μm. (**B**) GFP signal as a function of depth for 920 nm 2P excitation and 1320 nm 3P excitation. (**C**) Comparison of the GFP signal and THG signal as a function of depth at 1320 nm. (**D**) Lateral

*Figure 3 continued on next page*

*Figure 3 continued*

resolution measurement in the THG image captured at 200 μm depth. Lateral intensity profile measured along the white line (indicated by the orange arrow) is fitted by a Gaussian profile for the lateral resolution estimation (scale bar = 50 μm). (**E**) Cross-section images of the central complex (CC) ring neurons through the cuticle with 1320 nm 3P excitation (green). THG imaging visualizes the tracheal arbors (yellow). Arrows indicate different CC compartments that are identified (scale bars = 30 μm). (**F–G**) Lateral resolution measurements in 3P images captured at 56 μm depth. (**F**) The GFP fluorescence profile of CC ring neurons (green) and (**G**) the THG profile of surrounding trachea (yellow). Lateral intensity profiles measured along the white lines are fitted by Gaussian profiles for the lateral resolution estimation (scale bars = 20 μm).

The online version of this article includes the following source data and figure supplement(s) for figure 3:

**Source data 1.** Source data for plots *Figure 3B and C*.

**Figure supplement 1.** Comparing fluorescent signal attenuation with 2P and 3P excitation in cuticle-intact and cuticle-removed imaging preparations.

**Figure supplement 1—source data 1.** Source data for plots for *Figure 3—figure supplement 1C and D*.

**Figure supplement 2.** Structural imaging of central complex using the cuticle-intact and cuticle-removed imaging preparations.

**Figure supplement 2—source data 1.** Source data for plots for *Figure 3—figure supplement 2B–H*.

**Figure supplement 3.** Structural imaging of mushroom body neurons using the cuticle-intact and cuticle-removed imaging preparations.

**Figure supplement 3—source data 1.** Source data for plots for *Figure 3—figure supplement 3B–H*.

**Figure supplement 4.** Motion analysis during through-cuticle imaging.

**Figure supplement 4—source data 1.** Source data for plots for *Figure 4—figure supplement 1*.

to 2P excitation and was capable of imaging brain regions below the esophagus. We further quantified the effective attenuation length (EAL) for 2P and 3P excitation, and we found $EAL_{920nm}$ = 41.7 μm, $EAL_{1320nm}$ = 59.4 μm within depth 1–100 μm, and $EAL_{1320nm}$ = 91.7 μm within depth 100–180 μm (*Figure 3B*, *Figure 3—source data 1*). The third harmonic generation (THG) signal from the head cuticle and the trachea was also measured as a function of depth. THG signal can be used to measure the EAL ($EAL_{THG}$) (*Yildirim et al., 2019*). The $EAL_{THG}$ within the cuticle was much larger than the $EAL_{THG}$ inside the brain, once again demonstrating the high ballistic transmission of the 1320 nm laser light through the head cuticle (*Figure 3C*, *Figure 3—source data 1*). The full width at half maximum (FWHM) of the lateral brightness distribution at 200 μm below the surface of the cuticle was ~1.4 μm for tracheal branches captured by the THG signal (*Figure 3D*). Similarity in the attenuation lengths of THG and 3P fluorescence signal indicates that the labeling of membrane-targeted GFP is uniform across the brain, validating the use of the fluorescence signal when quantifying the EALs.

Cuticle-removed preparations are widely used in the fly neuroscience imaging studies (*Seelig et al., 2010*; *Simpson and Looger, 2018*; *Wang et al., 2003*). To directly compare the spatial resolution of cuticle-intact and cuticle-removed imaging preparations, we imaged the entire brain in flies expressing membrane-targeted GFP pan neuronally. We found that in the superficial layers of the fly brain (i.e. ~50 μm), through-cuticle 2P and 3P imaging generated images with similar signal-to-background ratio (SBR) to cuticle-removed preparation (*Figure 3—figure supplement 1A, B*).

We were able to distinguish the mushroom body and the central complex neuropils clearly in both imaging preparations. 3P generated images with better SBR compared to 2P in both cuticle-intact and cuticle-removed preparations. We measured the EAL for both cuticle-removed and cuticle-intact 2P/3P imaging and found that removing the cuticle and underlying tissues increased the EAL by ~1.5× (*Figure 3—figure supplement 1C, D*, *Figure 3—figure supplement 1—source data 1*). As the imaging depth increases, the image contrast decreases. The degradation of the image contrast in both 2P and 3P images is manifested by the change of the slope (*Akbari et al., 2021*; *LaViolette and Xu, 2021*) in the semilog plot of fluorescence signal versus depth (*Figure 3—figure supplement 1C, D*). Within ~100 μm depth, 2P imaging provided reasonable contrast

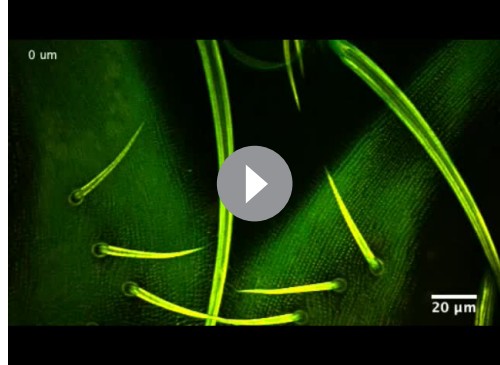

**Video 3.** Z stack of the ellipsoid body ring neurons expressing CD8-GFP. Imaging through the head cuticle at 1320 nm (3P) excitation (scale bar = 20 μm, semi compressed preparation).
https://elifesciences.org/articles/69094/figures#video3

when imaging through intact cuticle. While imaging is still possible with 3 P at the subesophageal zone ( >150 µm), it shows an increase in background that degrades image contrast when imaging beyond the esophagus (~150 µm). When cuticle was removed, 2 P imaging depth increased to ~180 µm, and 3P imaging depth increased to ~300 µm, reaching to the bottom of the fly brain. We further quantified and compared the laser power required to obtain the same fluorescence signal of 0.1 photon per laser pulse. For cuticle-removed fly, 3 P requires 1.4 nJ and 2 P requires 0.2 nJ on the brain surface to image the mushroom body. For cuticle-intact fly, 3 P requires 3.0 nJ and 2 P requires 0.5 nJ on the cuticle surface to image the mushroom body.

## 2P/3P imaging of mushroom body and central complex neurons through the fly head cuticle

To further test the performance of through-cuticle imaging with 2 P and 3 P excitation, we imaged the central complex ellipsoid body ring neurons. The insect central complex is a brain neuropil which processes sensory information and guides a diverse set of behavioral responses (*Pfeiffer and Homberg, 2014*; *Seelig and Jayaraman, 2015*; *Wolff et al., 2015*). It is composed of anatomically distinct compartments: the protocerebral bridge, ellipsoid body, fan-shaped body, and the noduli (*Wolff et al., 2015*). The ellipsoid body consists of a group of neurons, the ring neurons, that extend their axons to the midline forming a ring-like structure (*Pfeiffer and Homberg, 2014*; *Wolff et al., 2015*; *Xie et al., 2017*). Using an ellipsoid body-specific promoter, we expressed a membrane-targeted GFP in the ring neurons and imaged them with 2 P and 3 P excitation. Compared to 3 P (*Figure 3E*, *Video 3*), the resolution and contrast of images taken by 2 P was reduced when imaging through the cuticle at this depth (*Figure 3—figure supplement 2E*). Using ring neuron arbors and tracheal branches, we estimated the lateral resolution of the 3 P images. The FWHM of the lateral brightness distribution measured by a ring neuron's neurite cross-section was ~1.2 µm for the fluorescent signal (*Figure 3F*) and ~0.8 µm for tracheal branches captured by the THG signal (*Figure 3G*).

Next, we investigated whether cellular and subcellular resolution is achievable using the cuticle-intact imaging preparation, and compared our results to cuticle-removed 2 P and 3 P imaging. For these experiments, we imaged the Kenyon cells and the ellipsoid body ring neurons expressing a membrane-targeted GFP. Our data showed that Kenyon cell bodies were visible with cuticle-intact 2 P and 3 P imaging (*Figure 3—figure supplement 3A* and E), while deeper ellipsoid body ring neurons were only clearly distinguishable with 3 P imaging (*Figure 3—figure supplement 2A* and E). Our measurements showed that the cuticle-removed imaging preparation generated images with ~1.5× better axial resolution compared to cuticle-intact imaging preparations for both 2 P and 3- P imaging (*Figure 3—figure supplements 2 and 3*, *Figure 3—figure supplement 2—source data 1*, *Figure 3—figure supplement 3—source data 1*). 3 P imaging showed the same axial resolution in the mushroom body (~50 µm) and central complex (~100 µm) while 2 P imaging showed a deterioration of axial resolution for deep imaging in the central complex (*Figure 3—figure supplements 2 and 3*). We also investigated imaging stability during through-cuticle imaging by tracking ellipsoid body cell bodies in flies walking on a ball (*Video 4*). We did not detect major changes in the fluorescence intensity during walking. The average motion measured was 1.3 µm, which is much smaller than the size of a fly neuron (~5 µm) (*Figure 3—figure supplement 4*. *Figure 3—figure supplement 4—source data 1 Figure 3—figure supplement 4—source data 1*). Based on our results, we concluded that motion is not an issue during through-cuticle imaging at depths we have investigated. Together our results demonstrate that although both 2 P and 3 P excitation can be used for through-cuticle imaging at the superficial layers of the fly brain such as the mushroom body, 3 P outperforms 2 P in deeper brain regions such as the central complex especially

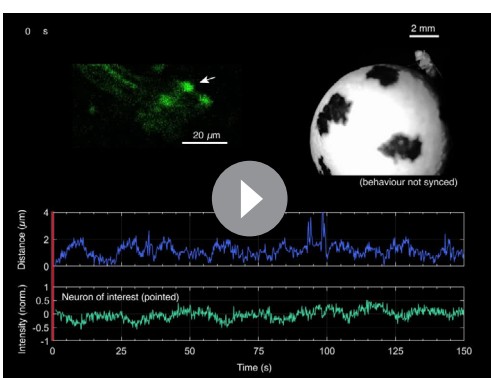

**Video 4.** T stack of the ellipsoid body ring neurons expressing CD8-GFP. Imaging through the intact head cuticle at 1320 nm (3P) excitation (scale bar = 20 µm, air sac-removed preparation).

https://elifesciences.org/articles/69094/figures#video4

when cellular and subcellular resolutions are necessary. This conclusion is consistent with the imaging studies conducted in the mouse brain (*Mok et al., 2019*; *Wang et al., 2018a*; *Wang et al., 2020*).

## 2P and 3P through-cuticle imaging does not induce heating damage to the fly brain tissue

The recommended power level for 2 P imaging of fly neural activity with cuticle-removed preparations is ~15 mW (*Seelig et al., 2010*). However, it is not known what the safe power levels for 2 P and 3 P imaging are, when imaging through the cuticle. 3 P excitation can induce heating in the mouse brain at high laser powers (*Wang et al., 2018a*; *Wang et al., 2018b*; *Wang et al., 2020*). Therefore, we measured how heat generated by 2 P and 3 P excitation impacts the fly brain using HSP70 protein as a marker for cellular stress response (*Lindquist, 1980*; *Podgorski and Ranganathan, 2016*). We first tested whether HSP70 protein levels reflect heat-induced stress in the fly brain. Flies that were kept at room temperature had low levels of the HSP70 protein (*Figure 4—figure supplement 1A*). In contrast, placing flies in a 30°C incubator for 10 minutes caused a significant increase in HSP70 protein levels across the fly brain (*Figure 4—figure supplement 1B*). Next, we tested whether 2 P and 3 P excitation causes an elevation in HSP70 protein levels when imaging through the cuticle. Head-fixed flies in a semi compressed imaging preparation were imaged either with 3 P or 2 P excitation. Our results showed that there was no measurable heat-stress response detected by the HSP70 protein levels when flies were exposed to 2 P (920 nm) and 3 P (1320 nm) excitation at 15 mW for 24 min (four 6-min intervals, see methods for details) (*Figure 4—figure supplement 1C-E*). However, increasing laser power to 25 mW for 3 P caused a significant increase in HSP70 protein levels in the fly brain (*Figure 4—figure supplement 1*). These results suggest that 2 P and 3 P cuticle-intact imaging is safe at power levels below 15 mW, similar to power levels used for 2 P cuticle-removed imaging.

## Whole brain 2P and 3P imaging in response to electrical stimulation

Encouraged by our structural imaging results, we next tested the applicability of 2 P and 3 P microscopy to capture neural activity in the entire fly brain through the intact head cuticle. In these experiments, we used a mild electric shock (1 s, ~ 5 V) and recorded neural activity in flies expressing GCaMP6s pan neuronally. We imaged the entire fly brain using the cuticle-intact and cuticle-removed imaging preparations with 2 P and 3 P. As expected, electrical stimulation generated a neural response in all the region of interests (ROIs) recorded across different depths of the fly brain. 3 P cuticle-removed preparation allowed us to image down to ~250 µm deep (*Figure 4B*), while the depth limit for 3 P cuticle-intact imaging was ~120 µm (*Figure 4A*). Additionally, we found that $dF/F_o$ for 3 P cuticle-removed imaging preparation was between 0.2 and 0.7 and for cuticle-intact imaging preparation it was between 0.2 and 0.5 (*Figure 4A and B*). We repeated the depth and $dF/F_o$ analysis for 2 P cuticle-intact and cuticle removed imaging. The depth limit for 2 P functional imaging through the cuticle was ~65 µm, while cuticle-removed imaging allowed optical access to ~120 µm (*Figure 4C and D*, *Figure 4—source data 1*). The $dF/F_o$ for 2 P cuticle-removed imaging preparation ranged between 0.2 and 0.4, while in the cuticle-intact preparation it was between 0.2 and 0.3. These results suggested that the presence of the cuticle and the underlying tissue decreases 2 P and 3 P functional imaging depth in the brain by ~2× and reduces $dF/F_o$. Similar to structural imaging, 3 P outperforms 2 P at deeper regions of the fly brain when recording neural activity in both cuticle-intact and cuticle-removed imaging preparations.

## Simultaneous 2P and 3P imaging of odor responses from mushroom body γ-lobes

We next recorded neural responses in the fly brain through the intact cuticle using a more natural stimulus, food odor. In these experiments, a custom odor delivery system was used where flies were head fixed and standing on a polymer ball under the microscope (*Figure 5A and B*). We expressed GCaMP6s in the mushroom body Kenyon cells and stimulated the fly antenna with the food odor apple cider vinegar (*Figure 5C*). Using a multiphoton microscope, odor-evoked $Ca^{2+}$ responses of mushroom body γ-lobes were simultaneously captured with 2 P (920 nm) and 3 P (1320 nm) excitation using the temporal multiplexing technique (*Ouzounov et al., 2017*). A brief 3 second odor stimulus triggered a robust fluorescence increase in the mushroom body γ-lobes (*Figure 5F*). Based on dopaminergic innervation, γ-lobes can be subdivided into five anatomical compartments (*Cohn*

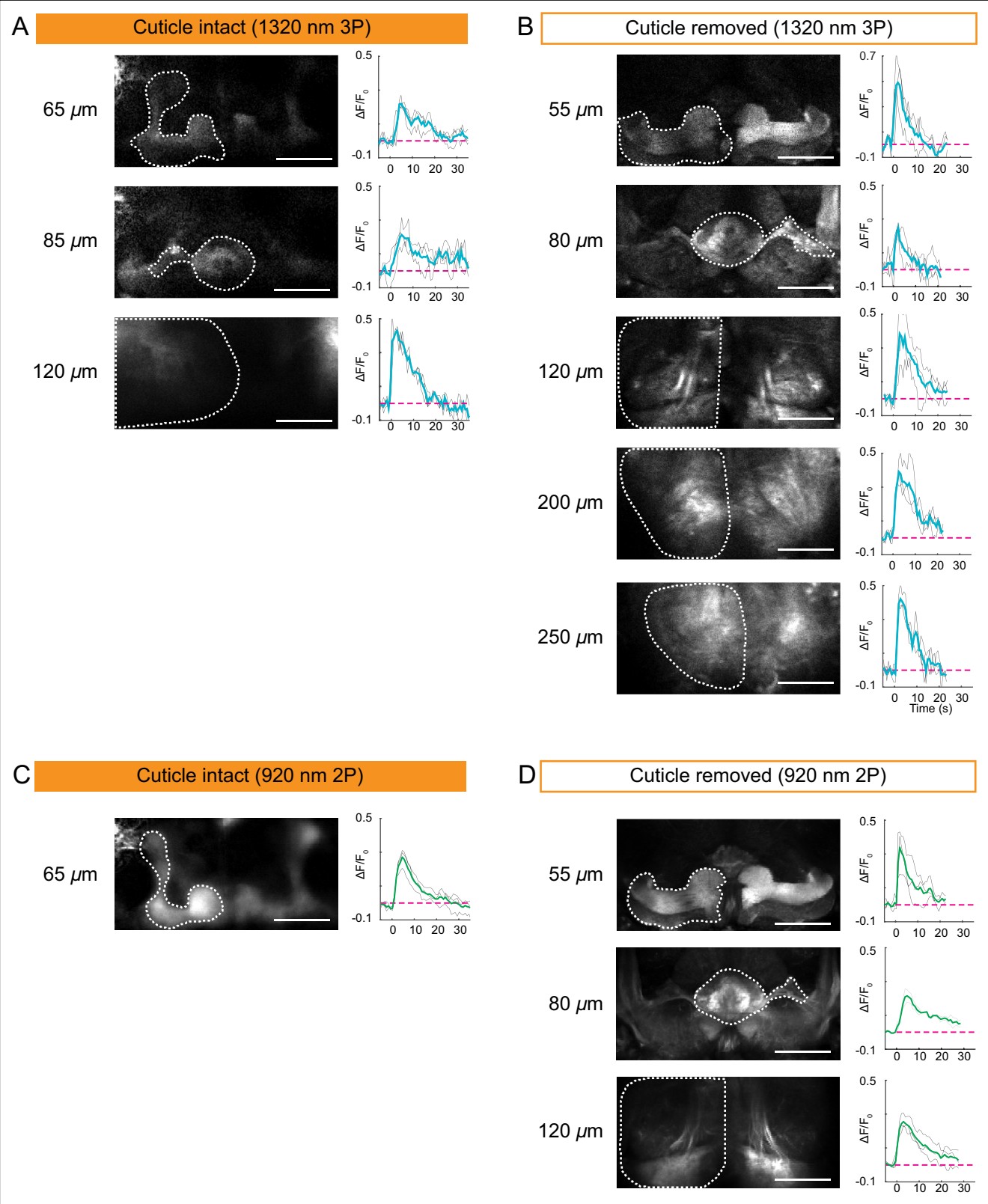

**Figure 4.** Cuticle-removed and cuticle-intact imaging of neural activity across the entire fly brain in response to electric shock. (**A–B**) 3P imaging of neural activity of the fly brain neuropil at indicated depths using (**A**) the cuticle-intact and (**B**) the cuticle-removed imaging preparations upon 1 s electrical stimulation. (**C–D**) 2P imaging of neural activity of the fly brain neuropil at indicated depths using (**C**) the cuticle-intact and (**D**) the cuticle-removed imaging preparations upon 1 s electrical stimulation. The cross-section images at different depths are shown on the left (scale bar = 50 $\mu$m).

*Figure 4 continued on next page*

*Figure 4 continued*

Activity traces within the ROIs enclosed by dotted white lines are shown on the right. Gray lines show the traces of individual stimulations and green lines show the traces of an average of three stimulations. Images were captured at 256 × 128 pixels/frame and 6.5 Hz frame rate for 3P and 113 Hz frame rate for 2P. 3P and 2P data were averaged to 1.1 Hz effective sampling rate for plotting.

The online version of this article includes the following source data and figure supplement(s) for figure 4:

**Source data 1.** Source data for plots for *Figure 4A–D*.

**Figure supplement 1.** HSP70 staining of fly brains after 2P and 3P imaging.

*et al., 2015*; *Figure 5D and E*). To investigate whether food odor is represented by different spatio-temporal patterns in the γ-lobe compartments, we calculated the normalized fluorescence signal for each compartment. No significant differences were observed in neural activity in responses to food odor stimulation across different compartments of the γ-lobes or between 2 P and 3 P excitation of GCaMP6s (*Figure 5G-I*, *Figure 5—figure supplement 1*, *Figure 5—source data 1*). We also recorded neural activity from Kenyon cell bodies in response to olfactory stimulation using 3 P excitation (*Figure 5—figure supplement 2*). Our data demonstrated that both 2 P and 3 P excitation can be used to image odor responses from mushroom body γ-lobes using through-cuticle imaging but for cell body imaging 3 P excitation is preferred.

## 2P through-cuticle imaging captures odor-evoked responses in behaving flies

To investigate how head compression impacts fly behavior and neural activity, we investigated how flies that are head compressed but allowed to walk on a spherical treadmill respond to an odor stimulation. Using our custom behavior/imaging setup (*Figure 6A*), we stimulated the fly antennae with food odor (apple cider vinegar), while recording neural activity from the mushroom body γ-lobes using 2 P excitation (920 nm) through the head cuticle. In these experiments, we also captured fly's behavioral responses using a camera that is synchronized with the 2 P microscope. A head-fixed fly was continuously exposed to a low speed air flow before and after the 3 s odor stimulus with the same air flow speed, and the behavioral responses of flies were captured by tracking the spherical treadmill motion using the FicTrac software during each trial (*Video 5*). Because internal states impact behavioral responses to food odors (*Lin et al., 2019*; *Sayin et al., 2019*), we used flies that are 24-hr food deprived. Previous studies have demonstrated that during food odor exposure, hungry flies increase their walking speed, orient, and walk toward the odor stimulus. After odor stimulation, however, flies increase their turning rate which resembles local search behavior. The odor offset responses persist for multiple seconds after the odor exposure (*Álvarez-Salvado et al., 2018*; *Sayin et al., 2019*). In our experiments with semi head-compressed flies, flies increase their turning rate upon brief stimulation with food odor apple cider vinegar (*Figure 6B*, *Figure 6—source data 1*). During these experiments, we were able to capture odor-evoked neural responses from all mushroom body γ-lobe compartments reliably (*Figure 6C*, *Figure 6—source data 1*).

We further analyzed the odor-evoked changes in fly walking behavior and showed that after the brief exposure to food odor stimulus, flies increased their forward walking speed and turning rate (*Figure 6E–G*, *Figure 2*). These responses lasted for multiple seconds (*Figure 6H–K*). Moreover, statistical analysis showed that there is a significant difference between the average forward and rotational speed values before and after the food odor exposure (*Figure 6I and K*, *Figure 2*). Our results are in agreement with previous studies that quantified odor-induced changes in walking behavior in head-fixed flies (*Sayin et al., 2019*). Altogether, these results indicate that head-compressed flies in our spherical treadmill setup can walk and exhibit behavioral and neural responses to odor stimulation.

## 2P through-cuticle imaging captures chronic odor-evoked responses

Studying how neural circuits change activity during learning or in alternating behavioral states requires chronic imaging methods that permit recording neural activity over long time scales. Leveraging our preparation, we pushed the limits of functional imaging of the fly brain in response to food odor stimulation at longer time scales (12 hr). Using a custom odor delivery system, we stimulated the fly antenna with food odor (apple cider vinegar) every 4 hr while imaging through the head cuticle using 2 P excitation (920 nm) (*Figure 7A*, *Video 6*). We calculated the normalized peak fluorescent signal

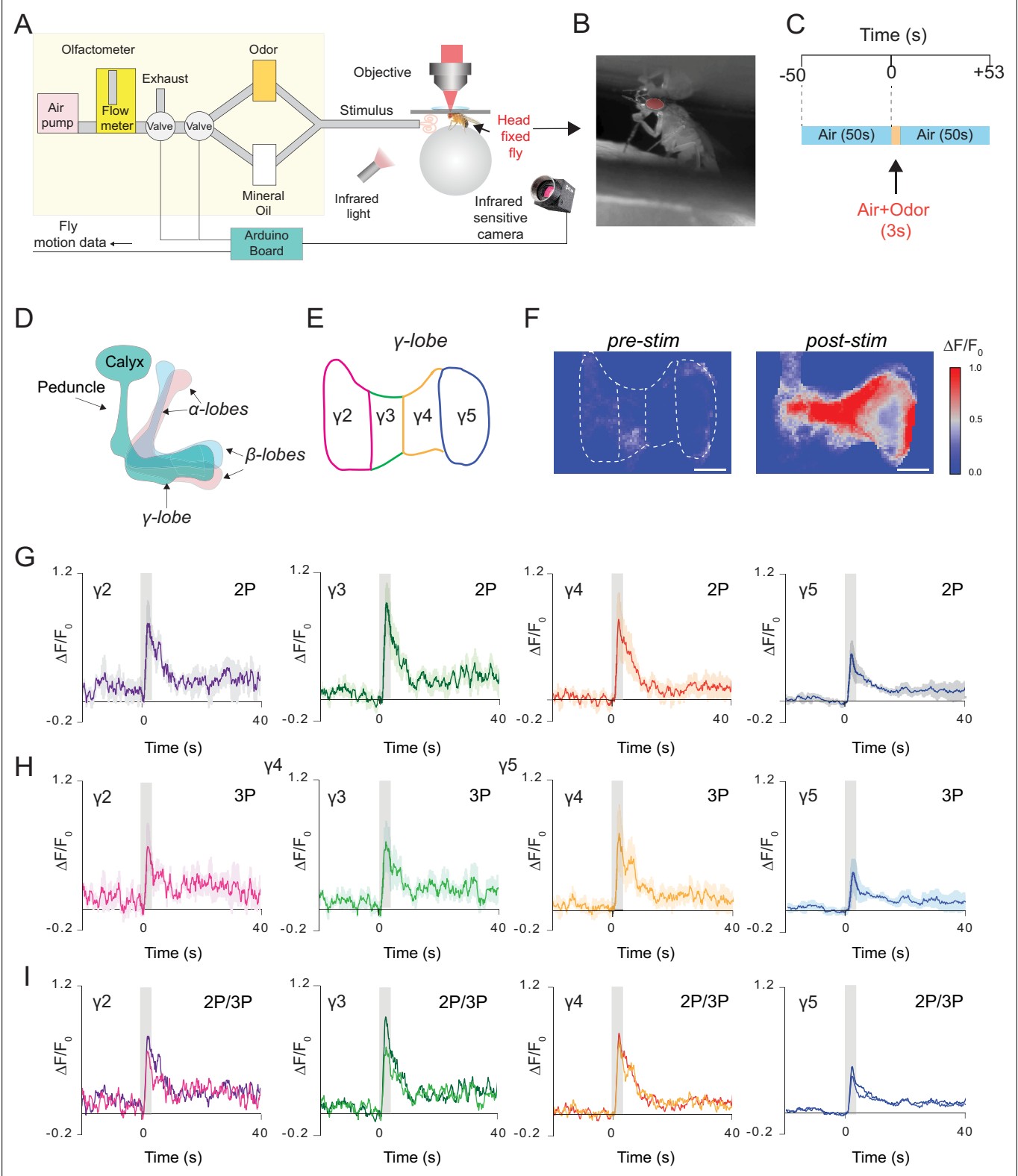

**Figure 5.** Simultaneous 2P and 3P functional imaging of short-term odor-evoked responses of the mushroom body Kenyon cells. (**A**) Schematic of the custom-made olfactometer and the through-cuticle functional imaging setup. (**B**) Picture of the head-fixed fly on the ball under the multiphoton microscope. (**C**) Stimulus timeline. The same stimulus scheme was repeated five times using the same odor. (**D**) Schematic of the mushroom body anatomy indicating the locations of (α, β and γ) lobes. (**E**) γ-Lobes have discrete anatomical compartments (shown as γ2–γ5). (**F**) GCaMP6s is expressed in the mushroom body Kenyon cells. Normalized (ΔF/F₀) GCaMP6s signal is shown before (left) and after (right) odor stimulus (scale bar = 20 m). (**G**)

*Figure 5 continued on next page*

*Figure 5 continued*

Odor-evoked responses of Kenyon cells captured by 2P excitation at 920 nm and (**H**) 3P excitation at 1320 nm. (**I**) Comparison of the average responses captured by simultaneous 2P and 3P imaging over time (n = 3 flies, 4–5 trials per fly, data are presented as mean ± SEM in (**G**) and (**H**), gray bar indicates when stimulus is present). Average laser powers are 5 mW at 920 nm and 4 mW at 1320 nm. Images were captured at 160 × 165 pixels/frame and 13.2 Hz frame rate. 2P and 3P data were averaged to 6.8 Hz effective sampling rate for plotting.

The online version of this article includes the following source data and figure supplement(s) for figure 5:

**Source data 1.** Source data for plots *Figure 5G-I*.

**Figure supplement 1.** Single-trial neural activity traces of odor-evoked responses in the mushroom body γ5 lobes.

**Figure supplement 2.** Neural activity traces from individual Kenyon cell bodies.

per fly in each γ-lobe compartment and time point as a metric representing the food odor response strength during chronic imaging (*Figure 7B–E*, *Figure 7—source data 1*; *Figure 7—source data 2*; *Figure 7—source data 3*; *Figure 7—source data 4*). Our analysis showed that the odor-evoked neural responses did not change with food and water deprivation in any of the γ-lobe compartments imaged (*Figure 7*, *Figure 7—source data 1*). During these long-term imaging experiments, we captured the fly's behavior in parallel with the odor stimulation to assure that the fly stayed alive during long-term imaging (*Video 6*). These results suggest that the cuticle-intact imaging method developed here allows recording of neural activity within an individual fly over long time scales (12 hr), which was previously not possible with commonly used cuticle-removed imaging preparations.

## Discussion

Imaging through the fly cuticle was considered to be not feasible at the wavelengths typically used for 2 P ( ~ 920 nm) and 3 P ( ~ 1300 nm) imaging because of concerns about cuticle absorption (*Lin et al., 2015*; *Tao et al., 2017*). By quantitatively measuring the optical properties of the fly cuticle at wavelengths that correspond to 2 P and 3 P imaging, we discovered that fly cuticle transmits long wavelength light with surprisingly high efficiency (*Figure 1*). We found that it is not the absorption by the cuticle but rather the opacity of the air sacs and the tissues located between the head cuticle and the brain that limit the penetration depth of multiphoton imaging (*Video 2*). By compressing the fly head using a glass coverslip, we reduced the volume of the air sacs between the cuticle and the brain, which increases the transmission of laser light and therefore allows high resolution imaging of the fly brain through the intact cuticle (*Figure 2*). Careful assessments showed that semi head compression does not cause measurable differences in fly courtship (*Figure 2—figure supplement 1A*), or olfactory behaviors (*Figure 6*). Our results clearly demonstrate that long excitation wavelength (e.g. ~1700 nm) is not necessary for imaging the fly brain through the cuticle and our fly preparations enable cuticle-intact 2 P and 3 P imaging of common fluorophores (e.g. GFP and GCaMPs) at 920 nm and 1320 nm, respectively (*Figures 2–7*). While we did not see noticeable differences in the recorded activity traces when performing simultaneous 2 P and 3 P functional imaging of the mushroom body, 3 P imaging has a better SBR than 2 P imaging in deeper regions of the fly brain such as the central complex.

Investigating how physiological states, sleep, and learning change the function of neural circuits requires tracking the activity of molecularly defined sets of neurons over long time scales. These experiments require long-term imaging methods to record neural activity in vivo. The through-cuticle imaging method developed here significantly extends the time frame of current in vivo imaging preparations used for anatomical and functional studies in fly neuroscience. Our imaging method will allow researchers to capture the activity of neural populations during changing behavioral states; facilitate decoding of neural plasticity during memory formation; and might permit observation of changes in brain structures during development and aging. Our first demonstration of long-term functional imaging of the fly brain captures food odor responses from mushroom body γ-lobes for up to 12 hr continuously. Our results suggest that odor-evoked $Ca^{2+}$ responses did not change during the repeated odor stimulation. Even longer imaging time is possible by feeding flies under the microscope. We performed 2 P imaging for demonstrating the possibility of long-term recording of neural activity because conventional 2 P microscopy has adequate penetration depth for imaging the behavioral responses within the mushroom body, and 2 P microscopy is widely used by the fly neuroscience community. On the other hand, our deep functional imaging data (*Figure 4*) showed that combination

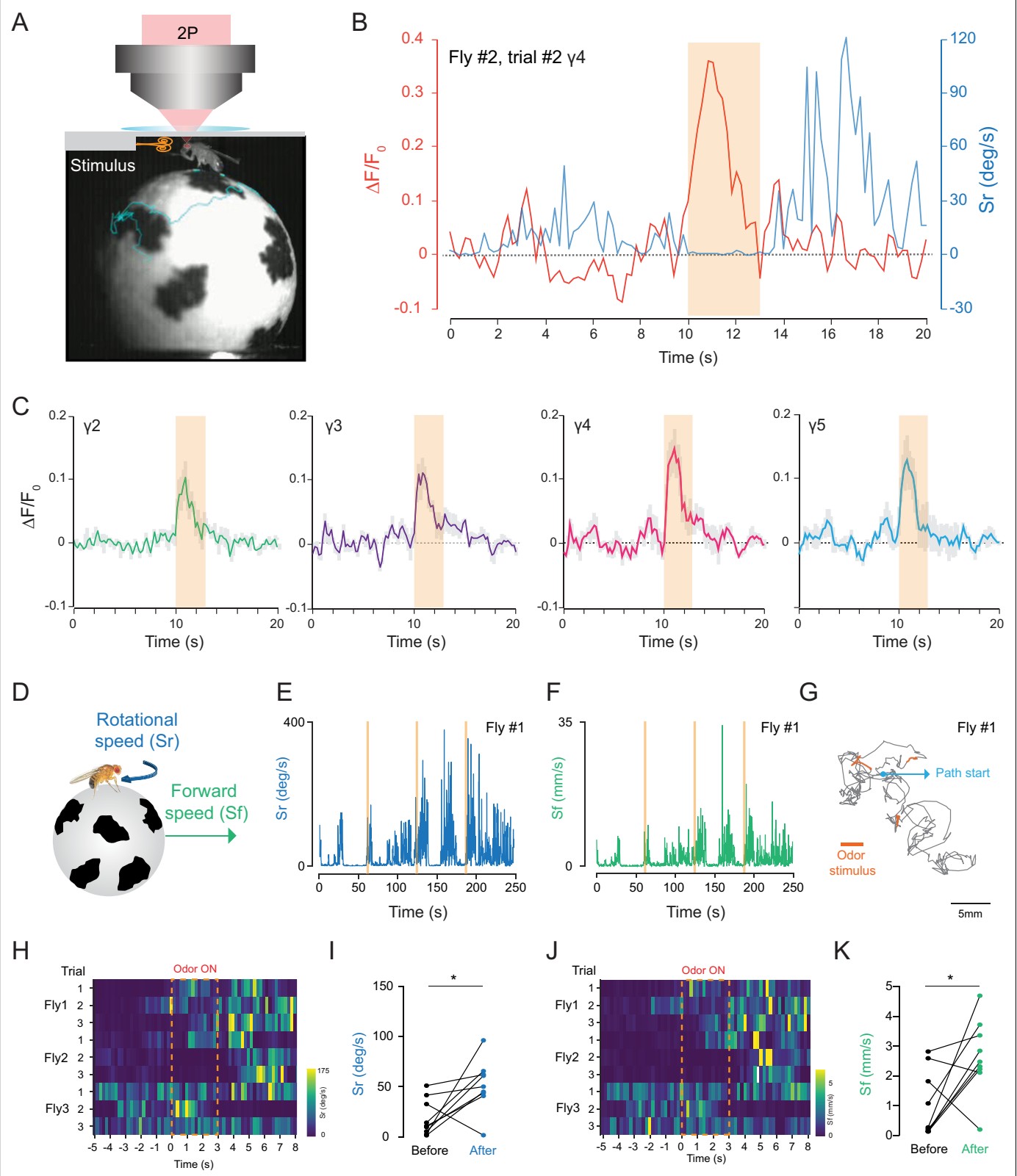

**Figure 6.** 2P functional imaging of odor-evoked responses in walking flies. (**A**) Schematic of the custom-made odor delivery and spherical treadmill system. (**B**) Odor-evoked response in mushroom body four compartment is overlaid with the rotational speed (Sr) measured at the same time. (**C**) Normalized (ΔF/F$_0$) GCaMP6s signal is shown during the odor stimulation experiments. Odor-evoked responses of Kenyon cells are captured by 2P excitation at 920 nm (n = 3 flies, 3 trials per fly, data are presented as mean ± SEM). (**D**) Schematics showing measurements of the rotational and forward

*Figure 6 continued on next page*

*Figure 6 continued*

speed of flies on the spherical treadmill. (**E–G**) Representative plots for a single fly during the odor stimulation experiments showing rotational speed (Sr) (**E**), forward speed (Sf) (**F**) as a function of time, and the total calculated 2D fictive path (**G**). (**H–K**) Summary heatmap plots and statistical comparison for rotational (**H–I**) and forward speed (**J–K**) 5 s before and after the odor stimulation (n = 3 flies, 3 trials per fly, paired-two tail t-test, p=0.0225). Average laser power at 920 nm is <10 mW. Images were captured at 256 × 128 pixels/frame and 17 Hz frame rate. 2P data were averaged to 5 Hz effective sampling rate for plotting.

The online version of this article includes the following source data for figure 6:

**Source data 1.** Source data for plots *Figure 6B and C*.

**Source data 2.** Source data for plots *Figure 6E-K*.

of our cuticle-intact fly preparation and 3P imaging may provide the exciting possibility of long-term imaging in deeper regions of the fly brain such as the central complex. We note that the success rate of chronic imaging experiments was ~50% because of the drift in the axial position of the brain when imaging for long periods of time. Further optimizations might improve the success rate of chronic imaging. Our focus here was to develop cuticle-intact in vivo structural and functional imaging methods that can extend imaging quality and length for the fly brain. We anticipate that there will be a wide variety of uses for this technology in *Drosophila* neuroscience research.

# Materials and methods

**Key resources table**

| Reagent type (species) or resource | Designation | Source or reference | Identifiers | Additional information |
|---|---|---|---|---|
| Genetic reagent (*Drosophila melanogaster*) | Mef2-GAL4 | Bloomington *Drosophila* Stock Center | BDSC: 50,742 | |
| Genetic reagent (*Drosophila melanogaster*) | GMR15B07-GAL4 | Bloomington *Drosophila* Stock Center | BDSC: 48,678 | |
| Genetic reagent (*Drosophila melanogaster*) | GMR57C10-GAL4 | Bloomington *Drosophila* Stock Center | BDSC: 39,171 | |
| Genetic reagent (*Drosophila melanogaster*) | 10XUAS-IVS-mCD8-GFP | Bloomington *Drosophila* Stock Center | BDSC: 32,186 | |
| Genetic reagent (*Drosophila melanogaster*) | 20XUAS-IVS-GCaMP6s | Bloomington *Drosophila* Stock Center | BDSC: 42,746 | |
| Antibody | anti-GFP (Rabbit polyclonal) | Torrey Pines | TP40 | IF (1:1000) |
| Antibody | anti-HSP70 (Rat monoclonal) | Sigma | SAB5200204 | IF (1:200) |
| Antibody | anti-BRP (Mouse monoclonal) | DSHB | nc82 | IF (1:20) |
| Antibody | DyLight 488 (Goat polyclonal anti-Rabbit) | Invitrogen | 35,552 | IF (1:1000) |
| Antibody | AlexaFluor 546 (Goat polyclonal anti-Rat) | Invitrogen | A-11081 | IF (1:1000) |
| Antibody | AlexaFluor 633 (Goat polyclonal anti-Mouse) | Invitrogen | A-21052 | IF (1:250) |
| Other | PBS | Lonza BioWhittaker | #17-517Q | |
| Other | Vectashield | Vector Labs | #H-1000–10 | |

## Fly stocks

Flies were maintained on conventional cornmeal-agar-molasses medium at 23–25°C and 60–70% relative humidity, under a 12 hr light: 12 hr dark cycle (lights on at 9 A.M.).

*Figure 1*
Males, w[1118]/Y; 20XUAS-IVS-GCaMP6s; Mef2-GAL4.
*Figure 2*
Males, w[1118]/Y; 10XUAS-IVS-mCD8-GFP; Mef2-GAL4.

*Figure 2—figure supplement 1*
<u>Panel A:</u> Females and Males, Canton-S.
<u>Panel B–D:</u> Males, w[1118]/Y; 20XUAS-IVS-GCaMP6s; Mef2-GAL4.
*Figure 3*
<u>Panel A–D:</u> Males, w[1118]/Y; 10XUAS-IVS-mCD8-GFP; GMR57C10.
<u>Panel E–G:</u> Males, w[1118]/Y; 10XUAS-IVS-mCD8-GFP; GMR15B07-GAL4.
*Figure 3—figure supplement 1*
Males, w[1118]/Y; 10XUAS-IVS-mCD8-GFP; GMR57C10-GAL4.
*Figure 3—figure supplement 2*
Males, w[1118]/Y; 10XUAS-IVS-mCD8-GFP; GMR15B07-GAL4.
*Figure 3—figure supplement 3*
Males, w[1118]/Y; 10XUAS-IVS-mCD8-GFP; Mef2-GAL4.
*Figure 3—figure supplement 4*
Males, w[1118]/Y; 10XUAS-IVS-mCD8-GFP; GMR15B07-GAL4.
Males, w[1118]/Y; 20XUAS-IVS-GCaMP6s; GMR57C10-GAL4.
*Figure 4—figure supplement 1*
<u>Panel A, B, F:</u> Males, w[1118]/Y; 20XUAS-IVS-GCaMP6s; Mef2-GAL4.
<u>Panel C-E:</u> Males, w[1118]/Y; 10XUAS-IVS-mCD8-GFP; Mef2-GAL4.
*Figures 5–7, Figure 5—figure supplements 1–2*
Males, w[1118]/Y; 20XUAS-IVS-GCaMP6s; Mef2-GAL4.

## Optical transmission measurements of the fly head cuticle

The measurement setup and procedures are similar to our previous work (*Mok et al., 2021*). *Drosophila* cuticle was dissected from the dorsal head capsule of flies that are age and gender controlled (male, 5 days old). The dissected cuticle was sandwiched between two #1 coverslips (VWR #1 16004–094) with ~10 µL of UV curable resin (Bondic UV glue #SK8024) to avoid dehydration of the sample (*Figure 1A*). Measurements from each dissected cuticle was done within a day. The first several measurements were repeated at the end of all measurements to ensure that dehydration or protein degradation, which may affect the optical properties of the tissue, did not happen as the experiment progressed. The total transmission and ballistic transmission of cuticle samples were measured using a custom-built device (*Figure 1B*). For ballistic transmission experiments, light from a single-mode fiber was magnified and focused on the cuticle with a ~25 µm spot size. We assume that collimated light passes through the sample since the Rayleigh range for a 25 µm (1/e²) focus spot is approximately 0.8–1.3 mm in water (refractive index 1.33) for wavelengths between 852 nm and 1624 nm, which is much larger than the thickness of the entire coverslip sandwich preparation ( <400 µm). The transmitted light from the cuticle was then coupled to another single-mode fiber with identical focusing optics and detected with a power meter (S146C, Thorlabs). Such a confocal setup ensures that only the ballistic transmission is measured. The incident power is ~10 mW on the cuticle for each measurement. An InGaAs camera (WiDy SWIR 640 U-S, NiT) and a CMOS camera (DCC1645C, Thorlabs) were used to image the sample and incident beam to ensure that the incident light spot is always on the cuticle and to avoid the dark pigments (usually at the edge of the cuticle), ocelli, and possible cracks introduced during dissection. The ballistic transmission of the cuticle was then calculated as the power ratio between the ballistic transmissions through the cuticle ($PT^{SMF}$) and the surrounding areas without the cuticle ($PT_0^{SMF}$), i.e., a reference transmission through areas containing only the UV curable resin, using *Equation 1*:

$$T_{ballistic} = \frac{PT^{SMF}}{PT_0^{SMF}} \tag{1}$$

For measuring the total transmission, light from a single-mode fiber was magnified and focused on the cuticle with a ~50 µm spot size. We again assume that collimated light passes through the sample since the Rayleigh range for a 50 µm (1/e²) spot size is 5–10 mm for wavelengths between 532 nm and 1624 nm. An IS power meter (S146C, Thorlabs) is placed immediately after the sample to measure the total transmission. The incident power on the sample is ~10 mW. The same cameras were used to visualize the light spot and the cuticle when the IS is removed. The total transmission of the cuticle was

then calculated as the optical power ratio between the transmissions through the cuticle ($PT^{IS}$) and the reference transmission through areas containing only the UV curable resin ($PT_0^{IS}$), both measured by the IS (*Equation 2*).

$$T_{total} = \frac{PT^{IS}}{PT_0^{IS}}$$

(2)

Data in *Figure 1D and G* are acquired by manually translating the sample orthogonal to the light path. For *Figure 1E and H*, the samples were translated with a motorized stage to acquire a spatially resolved transmission map. We collected data from several locations for each wavelength (ballistic transmission, n = 56 measurements across 5 cuticle samples; total transmission, n = 20 measurements across 4 samples). We then calculated the mean and the standard error across all measurements for ballistic or total transmission for the plots shown in *Figure 1D and G*, respectively.

## 2P/3P imaging preparations

### Through-cuticle imaging preparation with head compression

All animals used for imaging experiments were male flies with indicated genotypes kept at 25°C incubators and maintained on conventional cornmeal-agar-molasses medium. Flies used for chronic functional experiments were 2–7 days old, and flies used for short-term functional experiments were 1–4 days old. To perform through-cuticle brain imaging, flies were first head fixed in a 40 mm weigh dish (VWR#76299–236) with a hole made with forceps. A drop of UV curable resin (Liquid plastic welder, Bondic) was applied to the head and thorax, which was then cured with blue light (~470 nm) and fused to a cover glass. The fly antennae are ensured to be fully exposed after curing. Fly proboscises were immobilized with blue light curable resin to minimize head motion caused by muscle contractions. *Video 1* explains the imaging preparation.

### Through-cuticle imaging preparation with air sac removal

The dorsal head air sacs were repositioned to the posterior most portion of the head. This was done by deeply anesthetizing the flies on ice for ~5 min. The flies were placed into a modified pipette, allowing their head to stick out of the tip. Dental wax was wrapped around the head stabilizing it to the pipette. A sharpened glass capillary held in a micromanipulator was used to make a small incision just medial to the eye on the dorsal posterior area of the fly's head. A sharpened tungsten needle curved into a micro hook, held in a micro manipulator was inserted into the incision, and run just under the cuticle to hook the dorsal air sac. The hook was pulled to the rear of the head, bringing the air sacs with it. The hook was then manipulated to release the air sac. The procedure was repeated on the other side. The incisions were closed using a very small amount of UV curable resin over the incision site. The flies were then allowed to recover for 24 hr at 25 degrees on conventional food. Flies used in *Figure 4*, *Figure 2—figure supplement 1*, *Figure 3—figure supplement 2*, *Figure 3—figure supplement 4* went through the air sac removal surgery.

### Cuticle-removed imaging preparation

Flies were anesthetized on ice for ~1 min then placed into a holder made from a 0.02 mm thick carbon steel sheet with a small hole cut to allow the dorsal thorax and dorsal part of the head to protrude through the sheet. The flies were fixed to the imaging chamber using a UV curable resin (Bondic) around the perimeter of the hole in the sheet. Approximately 500 µl of adult artificial hemolymph was placed on the imaging chamber and the head cuticle was removed using a 20-gauge needle to cut along the medial perimeter of the eyes, the dorsal posterior extent of the head between the eyes, and just posterior to the

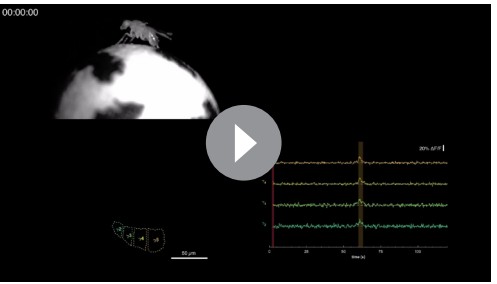

**Video 5.** Short-term 2P imaging of mushroom body γ-lobe neural activity captured through the intact fly head cuticle during walking and odor exposure. Functional imaging is performed in walking flies during a food odor stimulation (apple cider vinegar) with 2P excitation at 920 nm (semi compressed preparation, scale bar = 50 µm). Video is 5× speed up.
https://elifesciences.org/articles/69094/figures#video5

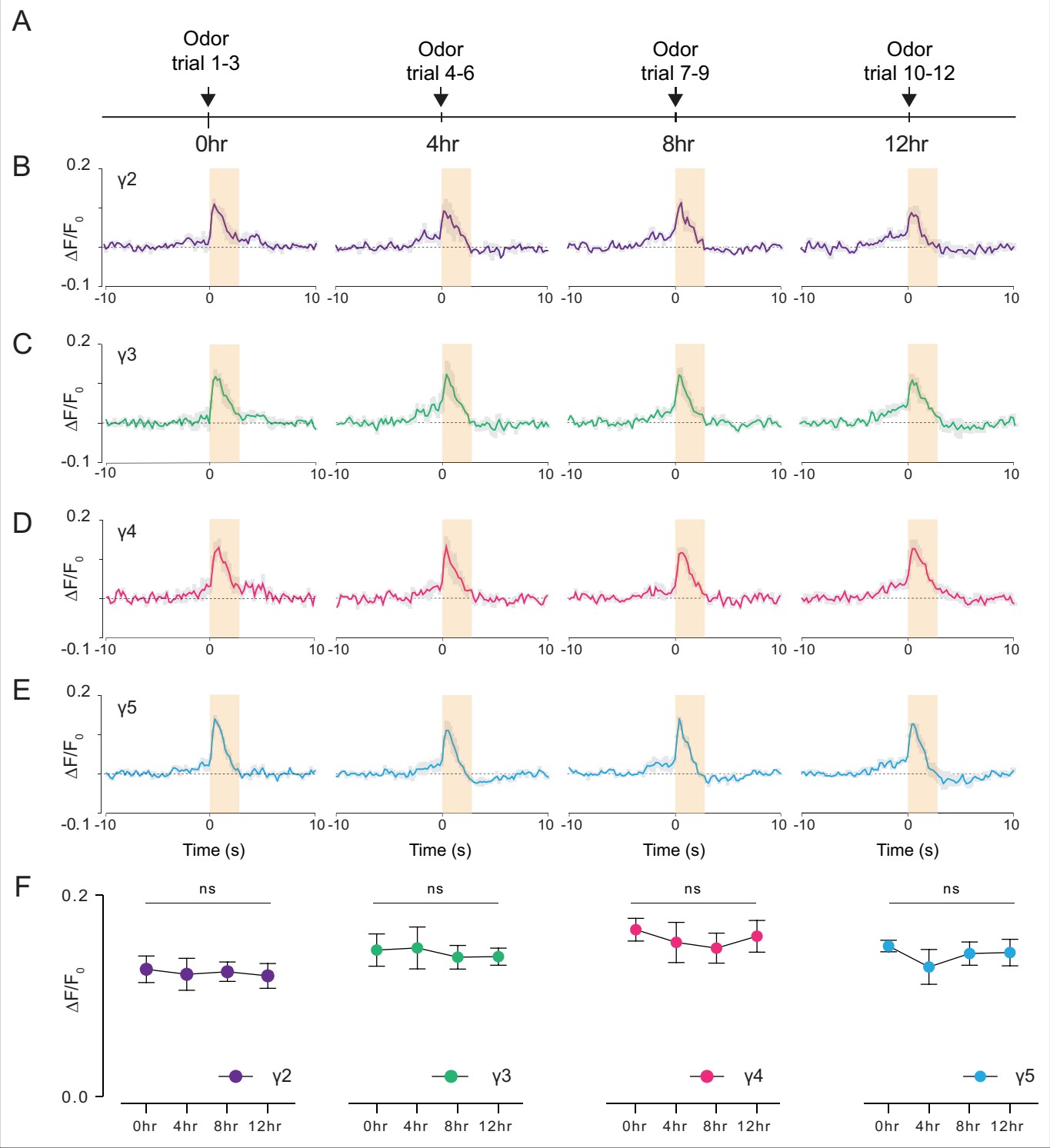

**Figure 7.** Long-term 2P imaging of odor-evoked responses of the mushroom body γ-lobes. (**A**) Stimulus timeline for long-term odor imaging. GCaMP6s fluorescence signal is captured from Kenyon cells axons innervating mushroom body γ-lobes using the semi compressed preparation. (**B–E**) Quantification of the normalized signal (ΔF/F$_0$) over time in each γ-lobe compartment. Light orange bar indicates when the odor stimulus is present. Each colored line indicates the average response of a fly over multiple trials in a given hour. The average response of three flies is shown. Each compartment's response is labeled with a different color. (**F**) Quantification of the peak amplitude across different time points and lobes (dF/F$_0$) (Two-way repeated measures ANOVA. Data are presented as mean ± SEM, ns, not significant; n = 3 flies, 3 trials per time point). Average laser power at 920 nm is <10 mW. Images were captured at 256 × 128 pixels/frame and 17 Hz frame rate. 2P data were averaged to 5 Hz effective sampling rate for plotting.

*Figure 7 continued on next page*

*Figure 7 continued*

The online version of this article includes the following source data for figure 7:

**Source data 1.** Source data for plots *Figure 7B-E* (t=0hr).

**Source data 2.** Source data for plot *Figure 7B-E* (t=4hr).

**Source data 3.** Source data for plots *Figure 7B-E* (t=8hr).

**Source data 4.** Source data for plots *Figure 7B-E* (t=12hr).

**Source data 5.** Source data for plot *Figure 7F*.

antenna along the front of the head. Any air sacs, fat bodies, or trachea on top of the exposed brain were removed with fine forceps.

## Multiphoton excitation source

### Whole brain 2P/3P imaging

The 3P excitation source is a wavelength-tunable optical parametric amplifier (NOPA, Spectra-Physics) pumped by a femtosecond laser (Spirit, Spectra-Physics) with a MOPA (Master Oscillator Power Amplifier) architecture. The center wavelength is set at 1320 nm. An SF11 prism pair (PS853, Thorlabs) is used for dispersion compensation in the system. The laser repetition rate is maintained at 333 kHz. The 2P excitation source is a Ti: Sapphire laser centered at 920 nm (Chameleon, Coherent). The laser repetition rate is at 80 MHz.

### 3P imaging

The excitation source is a wavelength-tunable optical parametric amplifier (OPA, Opera-F, Coherent) pumped by a femtosecond laser (Monaco, Coherent) with a MOPA architecture. The center wavelength is set at 1320 nm. An SF10 prism pair (10SF10, Newport) is used for dispersion compensation in the system.

### Simultaneous 2P/3P imaging and 2P imaging

The 3P excitation source is a wavelength-tunable optical parametric amplifier (NOPA, Spectra-Physics) pumped by a femtosecond laser (Spirit, Spectra-Physics) with a MOPA architecture. The center wavelength is set at 1320 nm. An SF11 prism pair (PS853, Thorlabs) is used for dispersion compensation in the system. The laser repetition rate is maintained at 400 kHz. The 2P excitation source is a Ti: Sapphire laser centered at 920 nm (Tsunami, Spectra-Physics). The laser repetition rate is at 80 MHz.

## Multiphoton microscopes

### Whole brain 2P/3P imaging

It is taken with a commercial multiphoton microscope with both 2P and 3P light path (Bergamo II, Thorlabs). A high numerical aperture (NA) water immersion microscope objective (Olympus XLPLN25XWMP2, 25 X, NA 1.05) is used. For GFP and THG imaging, fluorescence and THG signals are separated and directed to the detector by a 488 nm dichronic mirror (Di02-R488, Semrock) and 562 nm dichronic mirror (FF562-Di03). Then the GFP and THG signals are further filtered by a 525/50 nm band-pass filter (FF03-525/50, Semrock) and 447/60 nm (FF02-447/60, Semrock) band-pass filter, respectively. The signals are finally detected by GaAsP photomultiplier tubes (PMTs) (PMT2101, Thorlabs).

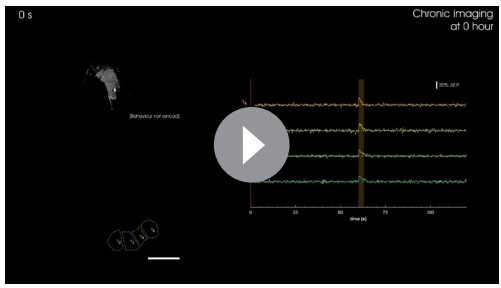

**Video 6.** Chronic 2P imaging of mushroom body γ-lobe neural activity captured through the intact fly head cuticle during odor exposure. Chronic functional imaging is performed during a food odor stimulation (apple cider vinegar) with 2P excitation at 920 nm (semicompressed preparation, scale bar = 50 μm). Video is 10× speed up.

https://elifesciences.org/articles/69094/figures#video6

### 3P imaging

A scan lens with 36 mm focal length (LSM03-BB, Thorlabs) and a tube lens with 200 mm focal length are used to conjugate the galvo mirrors to the back aperture of the objective. The same high NA water immersion microscope objective (Olympus XLPLN25XWMP2, 25 X, NA 1.05) is used. Two detection channels are used to collect the fluorescence signal and the THG signal by PMT with GaAsP photocathode (H7422-40, Hamamatsu). For 3-photon imaging of GFP and GCaMP6s at 1320 nm, fluorescence signal and THG signal were filtered by a 520/60 nm band-pass filter (FF01-520/60-25, Semrock) and a 435/40 nm band-pass filter (FF02-435/40-25, Semrock), respectively. For signal sampling, the PMT current is converted to voltage and low pass filter (200 kHz) by a transimpedance amplifier (C7319, Hamamatsu). Analog-to-digital conversion is performed by a data acquisition card (NI PCI-6110, National Instruments). ScanImage 5.4-Vidrio Technologies (*Pologruto et al., 2003*) running on MATLAB (MathWorks) is used to acquire images and control a movable objective microscope (MOM, Sutter Instrument Company).

### Simultaneous 2P/3P imaging and 2P imaging

A scan lens (SL50-3P, Thorlabs) and tube lens (TL200-3P, Thorlabs) are used to conjugate the galvo mirrors to the back aperture of the objective. The same objective (Olympus XLPLN25XWMP2, 25 X, NA 1.05) is used. Fluorescence signals are detected by PMT with GaAsP photocathode (H7422-40, Hamamatsu). For GFP and GCaMP6s imaging, fluorescence signal passes through a 466 nm dichronic mirror (Di02R-466) and filtered by a 520/60 nm band-pass filter (FF01-520/60-25, Semrock). For signal sampling, the PMT current is converted to voltage by a 10-MHz transimpedance amplifier (C9999, Hamamatsu). An additional 1.9-MHz low-pass filter (Minicircuts, BLP-1.9+) was used before digital sampling. Analog-to-digital conversion is performed by a data acquisition card (NI PCI- 6115, National Instruments). ScanImage 3.8 -Vidrio Technologies (*Pologruto et al., 2003*) running on MATLAB (Math-Works) was used to acquire images.

### Temporal multiplexing

Simultaneous imaging with 2P and 3P excitation is achieved by temporal multiplexing of the 920 nm Ti: Sapphire laser and 1320 nm Spirit-NOPA laser. The setup is similar to the one described in a previous study (*Ouzounov et al., 2017*). Briefly, two lasers were combined with a 980 nm long pass dichronic mirror (BLP01-980R-25, Semrock) and passed through the same microscope. They were spatially overlapped at the same focal position after the objective with a remote focusing module in the 2P light path. The 920 nm laser was intensity modulated with an electro-optic modulator (EOM), which was controlled by a transistor-transistor logic (TTL) waveform generated from a signal generator (33,210 A, Keysight) that is triggered by the Spirit-NOPA laser. The EOM has high transmission for 1 μs between two adjacent Spirit-NOPA laser pulses that are 2.5 μs apart. By recording the waveform from the signal generator and the PMT signal simultaneously, the 2P and 3P excited fluorescence signals can be temporally demultiplexed with postprocessing using a custom MATLAB script.

### Pulse energy comparisons to obtain 0.1 photon/pulse

The comparison follows the framework described in our previous work (*Wang et al., 2020*). In brief, a calibration factor that relates the pixel intensity of the image and the number of detected photons is first acquired by using a photon counter (SR400, Stanford instrument). Then, the brightest 0.25% pixel values of a frame from the whole brain stack are taken as the fluorescence signal and are converted to number of detected photons. Finally, the pulse energy required to obtain 0.1 photon/pulse can be calculated with measured power on the fly surface. The pulse width of the laser pulse to obtain a signal of 0.1 photon/pulse is normalized to 60 fs to account for the difference in pulse width between 3P (~60 fs) and 2P (~100 fs).

## Imaging depth, resolution, and motion quantifications

In all comparisons, signal strength and effective NA for 2P and 3P imaging were similar.

### Resolution quantifications

During the imaging session, the fly was placed on ice to reduce motion. For the mushroom body, 2P and 3P images were taken with a field-of-view (FOV) of 270 × 270 μm with a pixel count of 512 × 512.

The zoomed-in images were taken with an FOV of 75 × 75 μm with a pixel count of 512 × 512. For the central complex, 2P and 3P images were taken with an FOV of 250 × 126 μm with a pixel count of 512 × 256. The zoomed-in images were taken with an FOV of 50 × 50 μm with a pixel count of 256 × 256. A step size of 1 μm was taken for axial resolution measurement.

### Motion quantifications

Motion artifact during cuticle-intact in vivo 3P imaging was quantified by imaging ellipsoid body ring neurons expressing GFP. A video of 150 s is taken at a frame rate of 6.5 Hz with a field of view of 74 × 37 μm and pixel count of 256 × 128. The motion was calculated with the 'landmark' output that targets a single neuron from TurboReg plugin in ImageJ during image registration. After image registration, the intensity change of one neuron (I), as indicated in the ROI (*Figure 3—figure supplement 4*), in time was normalized according to the formula $(I - I_0)/I_0$. $I_0$ is taken as the mean of all intensity value (I) of the trace.

### Whole brain signal attenuation quantification

2P and 3P image stacks were taken with an FOV 200 × 100 μm with a pixel count of 512 × 256. Axial step sizes of 5 and 10 μm were used for cuticle-intact and cuticle-removed fly, respectively. The imaging power was increased with imaging depth to keep the signal level approximately constant. The maximum power on the fly brain was 15 mW for both 2P and 3P. The signal (S) of each frame was calculated as the average of the brightest 0.25% pixel values and then normalized by the imaging power (P) on the fly surface. The normalization was $S/P^2$ and $S/P^3$ for the 2P and 3P stacks, respectively. The EAL was then derived by least squares linear regression of the normalized fluorescence signal at different imaging depths.

## Electrical stimulation during 2P/3P functional imaging

### Cuticle-removed preparation

A tungsten wire was inserted into the adult artificial hemolymph on top of the exposed fly brain and secured in place with UV curable resin. A copper wire was placed in contact with ventral portion of the body of the fly and secured in place with UV curable resin. The wires were connected to a variable power supply with the tungsten positive side interrupted with a normally open relay module controlled with a microcontroller (Arduino Uno R3).

### Cuticle-intact preparation

Flies were prepared as described before for head compression imaging (*Video 1*). A 26-gauge copper wire was secured to the glass cover slip next to the head and another copper wire was secured to the ventral portion of the body. Low melting agarose (GeneMate #E-3126–25) with 0.5 M NaCl (Sigma #S7653-250G) was used to make an electrical connection between the wires and the fly, making sure the electrical path runs through the body.

### Electrical stimulation and imaging

Both the cuticle-intact and cuticle-removed flies were imaged with 2P and 3P excitation, taking images at different depths throughout the brain. For electrical stimulation the flies were stimulated for 1 s at 5 V and imaged at various depths to see a consistent GCaMP signal increase. 3P activity was taken with an FOV of 200 × 100 μm with pixel count of 256 × 128. The frame rate was 6.5 Hz. Every five frames were averaged to achieve an effective frame rate of 1.3 Hz. 2P activity was taken with an FOV of 200 × 100 μm with pixel count of 256 × 128. The frame rate is 113 Hz. Every 100 frames are averaged to achieve an effective frame rate of 1.1 Hz. ROIs were generated by manual segmentation. The baselines of the activity traces (F0) for each ROIs were determined using a rolling average of 4 s over the trace after excluding data points during electric stimulation. The activity traces (F) were normalized according to the formula $(F - F_0)/F_0$. Three stimulations were done for each depth.

## Olfactory imaging conditions and preparation of flies used in imaging experiments

### Simultaneous 2P/3P functional imaging

Flies were food deprived for 16–24 hr in vials with a wet Kim wipe. Each odor stimulation trial consisted of 50 s of clean mineral oil, 3 s of undiluted apple cider vinegar stimulus, and another 50 s of mineral oil. Between trials, scanning was stopped for 20 s to minimize the risk of imaging-induced tissue stress. Five trials were performed sequentially. Images were captured at 160 × 165 pixels/frame and 13.2 Hz frame rate. 2P and 3P data were averaged to 6.8 Hz effective sampling rate for plotting.

### 2P functional imaging in behaving flies

Flies were head fixed using a custom 3D-printed apparatus which also holds the tube for odor delivery. In this setup, flies are allowed to walk on a spherical treadmill and turn toward the odor stimuli. The odor stimulus is located on the right side of the fly. Each odor stimulation trial consisted of 60 s of clean mineral oil, 3 s of undiluted apple cider vinegar stimulus, and another 60 s of mineral oil. Every change of odor triggers the acquisition software to save in a new file. The images were captured at 256 × 128 pixel resolution and 17 Hz frame rate. Three trials were performed sequentially.

### 2P chronic functional imaging

Flies used in long-term functional imaging experiments were kept on regular fly food before the first trial to assure that they were satiated. Each odor stimulation trial consisted of 60 s of clean mineral oil, 3 s of undiluted apple cider vinegar stimulus, and another 60 s of mineral oil. Every change of odor triggers the acquisition software to save in a new file. The images were captured at 256 × 128 pixel resolution and 17 Hz frame rate. Three trials were performed sequentially, and the three-trial block was repeated every 4 hr. Between trial blocks, scanning was stopped, and air passing through the stimulation tube was redirected to the exhaust valve to prevent desiccation. To further prevent desiccation, flies were placed on a water-absorbing polymer bead.

## Olfactory stimulation

### Odor delivery during 2P/3P simultaneous functional imaging

Food odor, apple cider vinegar, was delivered using a custom-built olfactometer as described previously (*Raccuglia et al., 2016*). Clean room air was pumped (Active Aqua Air Pump, 4 Outlets, 6 W, 15 L/min) into the olfactometer, and the flow rate was regulated by a mass flow controller (Aalborg GFC17). Two Arduino controlled 3-way solenoid valves (3-Way Ported Style with Circuit Board Mounts, LFAA0503110H) controlled air flow. One valve delivered the odorized airstream either to an exhaust outlet or to the main air channel, while another valve directed air flow either to the stimulus or control channel. The stimulus channel contained a 50 ml glass vial containing undiluted apple cider vinegar (volume = 10 ml) (Wegmans), while the control channel contained a 50 ml glass vial containing mineral oil (volume = 10 ml). Flies were placed approximately 1 cm from a clear PVC output tube (OD = 1.3 mm, ID = 0.84 mm), which passed an air stream to the antennae (~1L/min). The odor stimulus latency was calculated before the experiments using a photo ionization detector (PID) (Aurora Scientific). We sampled odor delivery using the PID every 20 ms and found average latency to peak odor amplitude was <100 ms across 34 measurements. Flies were stimulated with air (50 s), before and after the odor stimulus (odor + air, 3 s). Same stimulus scheme was repeated three times.

### Odor delivery during spherical treadmill and chronic imaging

An air supported spherical treadmill setup was used to record fly walking behavior during multiphoton imaging. Male flies at 5–6 days post eclosion were anesthetized on ice for about 2 min and mounted to a coverslip with semicompression as described in *Video 1*. The cover slip was glued to a custom 3D-printed holder with an internal airway to deliver airflow along the underside of the coverslip directly onto the antenna without interfering with the air supported ball. The air duct was positioned 90 degrees to the right of the fly about 1 cm away. Clean room air was pumped (Hygger B07Y8CHXTL) into a mass flow meter set at 1 L/min (Aalborg GFC17). The regulated airflow was directed through an Arduino controlled three-way solenoid pinch valve (Masterflex UX-98302–42) using 1/16" ID tubing. The valve directed the airflow either through 50 ml glass vile containing 10 ml of undiluted apple cider

vinegar for the stimulus, or through a 50 ml glass vile containing 10 ml of mineral oil for the control. The latency from stimulus signal from the Arduino to odor molecules arriving at the fly's antenna was measured using a photo-ionization detector (Aurora Scientific) prior to the experiments and found to be <200 ms to peak stimulus.

## Fly behavior during olfactory stimulation coupled with 2P/3P imaging

The spherical treadmill was manufactured by custom milling with 6061 aluminum alloy. The treadmill has a concave surface at the end for placing the ball, which is supported by airflow. We fabricated foam balls (Last-A-Foam FR-7110, General Plastics, Burlington Way, WA USA) that are 10 mm in diameter using a ball-shaped file. We drew random patterns with black ink on the foam balls to provide a high-contrast surface for the ball tracking analysis. Fly behavior was videotaped from the side to capture any movement by a CCD camera (DCC1545M, Thorlabs) equipped with a machine vision camera lens (MVL6 × 12 Z, Thorlabs) and 950 nm long pass filter (FELH0950, Thorlabs). The acquisition frame rate for video recording was set to 8 Hz under IR light illumination at 970 nm (M970L4, Thorlabs). The stimulus signal from the Arduino is captured by NI-6009 (National Instrument) using a custom script written in MATLAB 2020b (Mathworks) to synchronize with the behavior video in data analysis.

## Male courtship assay

5–6 days of wild-type virgin female and male flies were collected right after eclosion and aged at 25°C for ~5 days. On the day of the courtship assay, control group males were placed on ice for 1–5 min, then placed in the imaging chamber without being head compressed or head fixed. They were allowed to recover for 5 hr at 25°C before getting tested in the courtship assay. Experimental group flies went through the entire head-compression and head-fixing procedure described in *Video 1*. These flies were removed from the imaging chamber after being head fixed and allowed to recover for 5 hr at 25°C before getting tested in the courtship assay. To quantify male courtship behavior, male and female flies were aspirated into a 1 cm courtship chamber and allowed to interact for 30 min. Courtship assays were recorded using a camera (FLIR Blackfly, BFS-U3-31S4M-C).

## Immunohistochemistry for brain tissue damage assessment

To investigate laser-induced stress in the fly brain, we exposed 4–6 days old male flies (MB > UAS GFP) to 2P laser at 920 nm with 15 mW power or to a 3P laser at 1320 nm with 15 mW power. Flies were prepared using the medium compression preparation described previously. Control flies were prepared the same way and kept in the dark at room temperature for the duration of the experimental procedure. Laser scanning was done in the same depth as the MB γ-lobes. Each scan lasted for 6 min. Flies were rested for 6 min until the next imaging session. Each fly was exposed to four imaging sessions. Once the experiment was completed, fly brains were dissected and stained with the HSP70 antibody. For the positive control group, flies were exposed to 30°C for 10 min in an incubator to induce HSP70 expression. For the negative control group, flies were kept at room temperature. Brains from each experimental and control groups were dissected in phosphate-buffered saline (PBS) and incubated in 4% paraformaldehyde in PBS for 20–30 min at room temperature on an orbital shaker. Tissues were washed three to four times over 1 hr in Phosphate-buffered saline (PBS, calcium- and magnesium-free; Lonza BioWhittaker #17-517Q) containing 0.1% Triton X-100 (PBT, Phosphate-buffered saline + Triton) at room temperature. Samples were blocked in 5% normal goat serum in PBT (NGS-PBT) for 1 hr and then incubated with primary antibodies diluted in NGS-PBT for 24 hr at 4°C. Primary antibodies used were anti-GFP (Torrey Pines, TP40, rabbit polyclonal, 1:1000), anti-BRP (DSHB, nc82, mouse monoclonal, 1:20), and anti-HSP70 (Sigma, SAB5200204, rat monoclonal, 1:200). The next day, samples were washed five to six times over 2 hr in PBT at room temperature and incubated with secondary antibodies (invitrogen) diluted in NGS-PBT for 24 hr at 4°C. On the third day, samples were washed four to six times over 2 hr in PBT at room temperature and mounted with VECTASHIELD Mounting Media (Vector Labs, Burlingame, CA, USA) using glass slides between two bridge glass coverslips. The samples were covered by a glass coverslip on top and sealed using clear nail polish. Images were acquired at 1024 × 1024 pixel resolution at ~1.7 µm intervals using an upright Zeiss LSM 880 laser scanning confocal microscope and Zeiss digital image processing software ZEN. The power, pinhole size and gain values were kept the same for all imaged brains during confocal microscopy.

## Image processing and data analysis

### Resolution measurements

We measured the lateral or axial brightness distribution of small features within the fly brains using either the GFP fluorescence signal (*Figure 3F*) or the THG signal (*Figure 3D and G*). Lateral intensity profiles measured along the white lines were fitted by a Gaussian profile for the estimation of the lateral resolution. Axial intensity profiles measured were fitted with a Lorentzian profile to the power of 2 and 3 for 3P and 2P, respectively. The FWHM of the profiles is shown in the figures.

### Measurement of excitation light attenuation in the fly brain

The image stack was taken with 5 µm step size in depth, and the imaging power was increased with imaging depth to keep the signal level approximately constant. The signal (S) of each frame was calculated as the average of the brightest 0.25% pixel values and then normalized by the imaging power (P) on the fly surface. The normalization is $S/P^2$ and $S/P^3$ for the 2P and 3P stacks, respectively. The EAL is then derived by least squares linear regression of the normalized fluorescence or THG signal at different imaging depths (*Figure 3B and C*).

### Image processing for structural imaging

TIFF stacks containing fluorescence and THG data were processed using Fiji, an open-source platform for biological image analysis (*Schindelin et al., 2012*). When necessary, stacks were registered using the TurboReg plugin.

### Multiplex 2P and 3P functional imaging

TIFF stacks containing fluorescence data were converted to 32 bits, and pixel values were left unscaled. Lateral movement of the sample in the image series, if any, was corrected by TurboReg plug-in in ImageJ. Images acquired during the multiplexed 2P-3P imaging sessions were first median filtered with a filter radius of 10 pixels to reduce high amplitude noise. To compute $\Delta F/F_0$ traces, γ-lobe ROIs were first manually selected using a custom Python script. $F_0$ was computed as the average of 10 frames preceding stimulus onset. The $F_0$ image was then subtracted from each frame, and the resulting image was divided by $F_0$. The resulting trace was then low pass filtered by a moving mean filter with a window size of eight frames. Data were analyzed using Python and plotted in Microsoft Excel. Peak $\Delta F/F_0$ was determined by the peak value within 20 frames after the odor delivery.

### 2 P functional imaging in behaving flies and chronic functional imaging

Lateral movement of the sample in the image series, if any, was corrected by TurboReg plug-in in ImageJ. A custom script written in MATLAB 2016b is used for all subsequent processing. Every four frames are averaged to achieve an effective frame rate of 4.25 Hz. ROIs were generated by manual segmentation of the mushroom bodies. The baselines of the activity traces ($F_0$) for each ROIs are determined using a rolling average of 4 s over the trace after excluding data points during odor stimulation. The activity traces (F) were normalized according to the formula $(F - F_0)/F_0$. The trace is finally resampled to 5 Hz with spline interpolation to compile with the timing in the motion tracking trace.

### Fly walking behavior analysis

Fly walking traces were obtained using the FicTrac (Fictive path Tracking) software as published previously (*Moore et al., 2014*). The ball rotation analysis was performed using the 'sphere_map_fn' function, which allows the use of a previously generated map of the ball to increase tracking accuracy. We post-processed the raw output generated by FicTrac. To calculate forward and rotational speeds, we used the delta rotation vectors for each axis. Then, we down-sampled raw data from 8 Hz to 5 Hz by averaging the values in the 200 ms time windows. The empty data points generated from down-sampling were linearly interpolated.

### Male courtship behavior analysis

The courtship videos were scored manually, and the time of copulation was recorded per pair.

## Statistics

Sample sizes used in this study were based on previous literature in the field. Experimenters were not blinded in most conditions as almost all data analysis were automated and done using a standardized computer code. All statistical analysis was performed using Prism 9 Software (GraphPad, version 9.0.2). Comparisons with one variable were first analyzed using one-way ANOVA followed by Tukey's multiple comparisons post hoc test. Comparisons with more than one variable were first analyzed using two-way ANOVA. Comparisons with repeated measures were analyzed using a paired t-test. We used pair-wise log-rank (Mantel-Cox) test to compare the copulation percentage curves in the male courtship assays. P values are indicated as follows: ****$p<0.0001$; ***$p<0.001$; **$p<0.01$; and *$p<0.05$. Plots labeled with different letters in each panel are significantly different from each other.

## Acknowledgements

We thank Joe Fetcho, Andy Bass, David Owald, and members of the Yapici Lab for comments on the manuscript. We acknowledge Bloomington *Drosophila* Stock Centre (NIH P40OD018537) and the Developmental Studies Hybridoma Bank (NICHD of the NIH, University of Iowa) for reagents. We thank Li Yan McCurdy (Yale University) and Matt Einhorn (Cornell University) for help with the design and construction of the custom built olfactometer, and Nancy M Bonini (University of Pennsylvania) for her advice on the HSP70 antibody. Research in NY's laboratory is supported by a Cornell University Nancy and Peter Meinig Family Investigator Program, a Pew Scholar Award, the Alfred P Sloan Foundation Award, AFAR Research Grant for Junior Faculty, NSF NeuroNex Program Grant (DBI-1707312), NIH R35 ESI-MIRA Grant (R35GM133698-01) and a Cornell Neurotech Mong fellowship.

## Additional information

### Funding

| Funder | Grant reference number | Author |
|---|---|---|
| National Science Foundation | DBI-1707312 | Nilay Yapici Chris Xu |
| National Institute of General Medical Sciences | R35 GM133698 | Nilay Yapici |
| Pew Charitable Trusts | Scholars Award | Nilay Yapici |
| Alfred P. Sloan Foundation | Scholars Award | Nilay Yapici |
| American Federation for Aging Research | Grants for Junior Faculty | Nilay Yapici |
| NSF NeuroNex Program | DBI-1707312 | Nilay Yapici |

The funders had no role in study design, data collection, and interpretation, or the decision to submit the work for publication.

### Author contributions

Max Jameson Aragon, Conceptualization, Data curation, Writing – original draft; Aaron T Mok, Conceptualization, Data curation, Writing - review and editing; Jamien Shea, Haein Kim, Nathan Barkdull, Data curation; Mengran Wang, Data curation, Writing – original draft; Chris Xu, Conceptualization, Funding acquisition, Supervision; Nilay Yapici, Conceptualization, Formal analysis, Funding acquisition, Project administration, Supervision, Writing – original draft, Writing - review and editing

### Author ORCIDs

Max Jameson Aragon ⓘ http://orcid.org/0000-0001-6935-0381
Aaron T Mok ⓘ http://orcid.org/0000-0003-0061-0258
Jamien Shea ⓘ http://orcid.org/0000-0002-6674-1165
Nathan Barkdull ⓘ http://orcid.org/0000-0002-1174-5046
Chris Xu ⓘ http://orcid.org/0000-0002-3493-6427
Nilay Yapici ⓘ http://orcid.org/0000-0002-1130-5083

Decision letter and Author response
Decision letter https://doi.org/10.7554/eLife.69094.sa1
Author response https://doi.org/10.7554/eLife.69094.sa2

## Additional files

### Supplementary files
• Transparent reporting form

### Data availability
All data generated or analyzed during this study are included in the manuscript and supporting file; Source Data files have been provided.

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
