## [Editor Report]

This study reports a way to record neural structures and activity in behaving *Drosophila* for up to 12 hours, without removing the cuticle. This opens the way to longer and more intact recordings, including post-imaging recovery of the flies. The authors describe their method for compression of air sacs to achieve this, and characterize image quality and responses using this method.

---

## [Decision Letter]

**Decision letter after peer review:**

Thank you for submitting your article "Multiphoton imaging of neural structure and activity in *Drosophila* through the intact cuticle" for consideration by *eLife*. Your article has been reviewed by 2 peer reviewers, one of whom is a member of our Board of Reviewing Editors, and the evaluation has been overseen by K VijayRaghavan as the Senior Editor. The reviewers have opted to remain anonymous.

Essential revisions:

The authors develop a non-invasive multiphoton imaging method to observe neural structure and activity in vivo in *Drosophila*. They achieve this by compression of air sacs to provide optical access to the brain while keeping the cuticle intact. The authors demonstrate calcium-sensor imaging from brain regions for up to 12 hours in behaving animals, which is a considerable improvement.

The reviewers were in agreement that this new method was potentially very useful, but it needed substantially more characterization to make a clear case for broad adoption. Specifically, the reviewers would like to see a characterization of the following:

1. Imaging spatial resolution and stability for still and functional imaging, especially for deeper structures.

2. A systematic comparison of resolution and performance for 2-photon, 3-photon, cuticle-removed, and the reported cuticle intact preparation.

3. Show feasibility for single-neuron and subcellular imaging, in dense and sparse expression systems.

In addition, the reviewers had a number of other points where better characterization and reporting would strengthen the paper.

*Reviewer #1:*

In this tools and resources paper, the authors demonstrate a way to record neuronal activity from flies without surgical removal of the head cuticle. It is valuable to be able to record from flies with minimal surgery. This means that the fly is less stressed, that the tissue is more healthy, and the recordings can go on for longer. The authors demonstrate optical recordings of odor responses through the cuticle, but not at the single-cell resolution that many readers would like. They also illustrate better imaging at greater depth using 3-photon illumination. The approach is potentially very useful, and the case for it would be even stronger were there a more complete comparison with conventional recordings with the cuticle removed.

1. Could the authors make a thorough comparison with cuticle-removed recordings? One would like to compare spatial resolution as well as signal-to-noise in the GCaMP recordings.

2. Could the authors obtain single neuron data? It would be useful to report the feasibility of the following:

A. To resolve single neurons optically in the midst of dense expression of the kind used in the paper in the mushroom body and ellipsoid body.

B. To express GCaMP in identified single neurons and see stimulus-triggered signals.

C. To sparsely express GCaMP in a population of neurons and see if one can isolate them.

D. To report if subcellular resolution can be obtained.

*Reviewer #2:*

The authors develop a non-invasive multiphoton imaging method to observe neural structure and activity in vivo in *Drosophila*. They achieve this by compression of air sacs to provide optical access to the brain while keeping the cuticle intact. Their results show that the through-cuticle preparation is compatible with 2- and 3-photon structural and functional imaging, with 3-photon imaging providing higher anatomic resolution in deeper brain regions. Through 2-photon calcium imaging of the mushroom body γ lobes, the authors demonstrate that odor-evoked responses can be studied short-term and long-term in behaving flies. The demonstration of the feasibility of functional imaging without surgery and of chronic imaging are true breakthroughs that will allow novel investigations of how the brain senses and interprets information over time to guide behavior. As we will describe in our feedback to the authors, additional experiments or analyses will allow the authors to demonstrate the strengths and limits of head compression and 3-photon imaging vis-a-vis different kinds of neurons and parts of the brain.

Major points to be addressed through new experiments:

1– The authors show that 3-photon imaging generates sharper anatomic pictures of neurons deep in the brain than does 2-photon imaging, but for functional imaging, only compare 3p versus 2p in a superficial structure (mushroom body γ lobes, Figure 4H). As the paper focuses on the utility of these techniques for functional imaging, the authors should provide data on the performance of 2- versus 3-photon functional imaging in a deeper brain region. It would also be nice to provide more detail of the spatial resolution of 3p in Figure 4, as only temporal traces are shown.

2 – The authors demonstrate that head compression allows chronic functional imaging through the cuticle, without surgery, in a single test case--the mushroom body axonal γ lobes. This is a great advance, but as the authors know, this is the single easiest thing they could image and is not characteristic of the rest of the fly brain. First, the lobes are quite superficial. Second, they contain hundreds of parallel Kenyon cell axons that respond to odor in a temporally and spatially correlated manner. (A) The authors need to put their compression technique through its paces by showing us whether it can or cannot be used for functional imaging of sparser neuronal populations in more tangled areas of the brain. (B) We also need to know the depth in the brain that is feasible for 2-photon imaging through the cuticle, and with respect to point 1 above, at what depth it becomes preferable to switch to 3-photon for functional imaging. (C) Finally, because the γ lobe compartments are large and thus stay in view even when the preparation is moving somewhat, it's not clear what sort of stability the authors obtain in this preparation. We need to know whether things like cell somata or isolated neurites can be stably resolved over time, or whether the fly's motion prevents this.

3 – It is wonderful that the authors can image for hours using this technique, and critical to the breakthrough to know the extent to which smashing the head causes brain damage. We suggest repeating the elegant heat shock protein analysis (Figure 4 supplement 1) on flies chronically imaged by 2p and 3p. We also suggest testing whether flies removed from the imaging stage/preparation can mate and produce offspring, as this would be a simple demonstration that they are still capable of sensory perception, decision-making, and complex motor programs.

In all the points above, we don't require a particular result for publication. Rather, because this is a methods-focused paper, we need a fuller picture of both the possibilities and limits of this technical approach--what it can do, and what it still cannot do.

---

## [Author Response]

Essential revisions:The authors develop a non-invasive multiphoton imaging method to observe neural structure and activity in vivo in *Drosophila*. They achieve this by compression of air sacs to provide optical access to the brain while keeping the cuticle intact. The authors demonstrate calcium-sensor imaging from brain regions for up to 12 hours in behaving animals, which is a considerable improvement.

We thank the reviewers for their kind words.

The reviewers were in agreement that this new method was potentially very useful, but it needed substantially more characterization to make a clear case for broad adoption. Specifically, the reviewers would like to see a characterization of the following:1. Imaging spatial resolution and stability for still and functional imaging, especially for deeper structures.

We have quantified axial resolution, which is a more sensitive measure of the optical resolution than the lateral resolution, in the cuticle-removed and cuticle-intact imaging preparations and tested imaging stability in cuticle-intact imaging preparations. New data and quantifications are now included in Figure 3—figure supplement 2, Figure 3—figure supplement 3, Figure 3—figure supplement 4 and Figure 4.

2. A systematic comparison of resolution and performance for 2-photon, 3-photon, cuticle-removed, and the reported cuticle intact preparation.

We have quantified the resolution and performance of 2P and 3P with cuticle-removed and cuticle-intact imaging preparations. New data and quantifications are now included in Figure 3—figure supplement 1, Figure 3—figure supplement 2, Figure 3—figure supplement 3 and Figure 4.

3. Show feasibility for single-neuron and subcellular imaging, in dense and sparse expression systems.

We have shown single-neuron imaging in dense (Kenyon cells) and sparse expression systems (Ellipsoid body ring neurons). We can obtain subcellular resolution for 2P in the Kenyon cells (cellular membrane and nucleus), and for 3P in the Kenyon cells (cellular membrane and nucleus) and ellipsoid body ring neurons (cellular membrane, nucleus and neuronal arbors) using the cuticle-intact imaging preparation. New data and quantifications are now included in Figure 3—figure supplement 2, Figure 3—figure supplement 3 and Figure 5 figure supplement 2.

In addition, the reviewers had a number of other points where better characterization and reporting would strengthen the paper.Reviewer #1:In this tools and resources paper, the authors demonstrate a way to record neuronal activity from flies without surgical removal of the head cuticle. It is valuable to be able to record from flies with minimal surgery. This means that the fly is less stressed, that the tissue is more healthy, and the recordings can go on for longer. The authors demonstrate optical recordings of odor responses through the cuticle, but not at the single-cell resolution that many readers would like. They also illustrate better imaging at greater depth using 3-photon illumination. The approach is potentially very useful, and the case for it would be even stronger were there a more complete comparison with conventional recordings with the cuticle removed.

We thank the reviewer for their kind words.

1. Could the authors make a thorough comparison with cuticle-removed recordings? One would like to compare spatial resolution as well as signal-to-noise in the GCaMP recordings.

We understand the reviewer’s point of view and agree that comparing spatial resolution obtained with cuticle-intact and cuticle-removed imaging preparations is important. We took the reviewers advice and quantified the spatial resolution in different regions of the fly brain using 2P and 3P imaging using the cuticle-removed and cuticle-intact preparations. We also imaged neural activity in response to electric shock in flies expressing GCaMP6s across the entire fly brain.

Our measurements showed that the cuticle-removed imaging preparation generated images with ~1.5X better axial resolution compared to cuticle-intact imaging preparations for both 2P and 3P imaging. These measurements were done using mushroom body Kenyon cells and central complex Ellipsoid body ring neurons. Signal attenuation for cuticle-intact and cuticle-removed preparations was also different for both 2P and 3P imaging. For cuticle-intact imaging 2P-EAL=45µm and 3P-EAL=61µm, while for cuticle-removed imaging 2P-EAL=72µm and 3P-EAL=99µm. These results indicate that cuticle intact imaging experiences a faster fluorescence signal decay for both 2P and 3P imaging. Based on the signal attenuation data, we estimated the imaging depth for 2P as 100µm and 3P as 150µm when imaging through intact cuticle. While imaging is still possible with 3P at the sub-esophageal zone (>150µm), it shows an increase in the background signals that degrades image contrast when imaging beyond the esophagus (~150µm). When cuticle is removed, imaging depth limits increased. For the cuticle-removed preparation, 2P imaging depth was increased to 180µm, and 3P imaging depth was increased to ~300µm, reaching the bottom of the fly brain. The new data and quantifications are shown in Figure 3—figure supplement 1, Figure 3—figure supplement 2 and Figure 3—figure supplement 3.

For the neural activity recordings using GCaMP6s, we found that deltaF/F_o_ for 2P cuticle-removed imaging prep ranged between 0.2 to 0.4 and for cuticle-intact prep it ranged between 0.2 to 0.3. On the other hand, deltaF/F_o_ for 3P cuticle-removed imaging prep ranged between 0.2 to 0.7, for cuticle-intact it ranged between 0.2 to 0.5. The new data and quantifications are shown in Figure 4.

2. Could the authors obtain single neuron data? It would be useful to report the feasibility of the following:A. To resolve single neurons optically in the midst of dense expression of the kind used in the paper in the mushroom body and ellipsoid body.

To demonstrate single-cell resolution, we imaged the Kenyon cells and ellipsoid body ring neurons and compared using cuticle-intact and cuticle-removed preparations. The new data is shown in Figure 3—figure supplement 2 and Figure 3—figure supplement 3. Our data demonstrates that Kenyon cell bodies are visible when imaging through the cuticle with both 2P and 3P excitation. However, ellipsoid body ring neurons are only clearly distinguishable with 3P excitation when imaging through the intact cuticle. The new results and measurements are added to the revised text.

B. To express GCaMP in identified single neurons and see stimulus-triggered signals.

To demonstrate single neuron GCaMP responses, we recorded neural activity from Kenyon cell bodies in response to olfactory stimulation using 3P (Figure 5—figure supplement 2).

We have also shown in our structural imaging experiments, we can identify single Kenyon cells with both 2P and 3P (Figure 3—figure supplement 3), although axial resolution is ~1.5X better for 3P compared to 2P. We have added a sentence to the text to indicate this information.

C. To sparsely express GCaMP in a population of neurons and see if one can isolate them.

In the revised version of the manuscript, we demonstrated subcellular resolution for Kenyon cells and Ellipsoid body ring neurons for 3P and for Kenyon cells only for 2P using our cuticle-intact fly preparation. GFP and GCaMP excitation and emission spectrums are very similar hence, we expect there won’t be any differences in axial resolution for GFP and GCaMP. We also included a data set showing single cell GCaMP recordings from Kenyon cells in Figure 5—figure supplement 2.

D. To report if subcellular resolution can be obtained.

Based on our images and axial resolution measurements shown in Figure 3—figure supplement 2 and 3, we can obtain subcellular resolution for 2P in the Kenyon cells (cellular membrane and nucleus), and for 3P in the Kenyon cells (cellular membrane and nucleus) and ellipsoid body ring neurons (cellular membrane, nucleus and neuronal arbors) using the cuticle-intact imaging preparation.

Reviewer #2:The authors develop a non-invasive multiphoton imaging method to observe neural structure and activity in vivo in *Drosophila*. They achieve this by compression of air sacs to provide optical access to the brain while keeping the cuticle intact. Their results show that the through-cuticle preparation is compatible with 2- and 3-photon structural and functional imaging, with 3-photon imaging providing higher anatomic resolution in deeper brain regions. Through 2-photon calcium imaging of the mushroom body γ lobes, the authors demonstrate that odor-evoked responses can be studied short-term and long-term in behaving flies. The demonstration of the feasibility of functional imaging without surgery and of chronic imaging are true breakthroughs that will allow novel investigations of how the brain senses and interprets information over time to guide behavior. As we will describe in our feedback to the authors, additional experiments or analyses will allow the authors to demonstrate the strengths and limits of head compression and 3-photon imaging vis-a-vis different kinds of neurons and parts of the brain.Major points to be addressed through new experiments:1– The authors show that 3-photon imaging generates sharper anatomic pictures of neurons deep in the brain than does 2-photon imaging, but for functional imaging, only compare 3p versus 2p in a superficial structure (mushroom body γ lobes, Figure 4H). As the paper focuses on the utility of these techniques for functional imaging, the authors should provide data on the performance of 2- versus 3-photon functional imaging in a deeper brain region.

We understand the reviewer’s concerns. In our revised manuscript, we have conducted whole brain functional imaging experiments using cuticle-intact and cuticle-removed preparations with 2P and 3P excitation. New results are shown in Figure 4. Our data demonstrated that, similar to structural imaging, 3P outperforms 2P at deeper regions of the fly brain when imaging neural activity in both cuticle-intact and cuticle-removed imaging preparations. We have revised the text to explain our results.

It would also be nice to provide more detail of the spatial resolution of 3p in Figure 4, as only temporal traces are shown.

In the revised version, we have done extensive analysis of 2P and 3P imaging using both cuticle-intact and cuticle-removed preparations (Please also see response to Reviewer 1 Point). The new results are shown in Figure 3 —figure supplement 1 and 3.

2 – The authors demonstrate that head compression allows chronic functional imaging through the cuticle, without surgery, in a single test case--the mushroom body axonal γ lobes. This is a great advance, but as the authors know, this is the single easiest thing they could image and is not characteristic of the rest of the fly brain. First, the lobes are quite superficial. Second, they contain hundreds of parallel Kenyon cell axons that respond to odor in a temporally and spatially correlated manner. (A) The authors need to put their compression technique through its paces by showing us whether it can or cannot be used for functional imaging of sparser neuronal populations in more tangled areas of the brain.

In the revised manuscript, we have done new experiments and measurements to quantify the limits of cuticleintact imaging. For resolution measurements we have imaged Kenyon cell bodies and ellipsoid body ring neurons. These analyses were done using structural imaging since, it is not possible to always find a natural stimulus for neurons in the fly brain. In addition, we do not expect GFP and GCaMP to have different resolution properties with 2P or 3P imaging. We have also added Figure 5 —figure supplement 2 to show that functional imaging is possible at a single-cell level in a population of neurons.

(B) We also need to know the depth in the brain that is feasible for 2-photon imaging through the cuticle, and with respect to point 1 above, at what depth it becomes preferable to switch to 3-photon for functional imaging.

We have investigated the depth limit of 2P and 3P imaging. Our measurements suggests that the imaging depth for 2P is 100µm and 3P as 150µm when imaging through intact cuticle. When cuticle is removed, imaging depth limit increases. For the cuticle-removed preparation, 2P imaging depth is increased to 180µm, and 3P imaging depth is increased to ~300µm, reaching the bottom of the fly brain. So, one can argue, when imaging through the cuticle it is preferable to switch to 3P at~100µm.

(C) Finally, because the γ lobe compartments are large and thus stay in view even when the preparation is moving somewhat, it's not clear what sort of stability the authors obtain in this preparation. We need to know whether things like cell somata or isolated neurites can be stably resolved over time, or whether the fly's motion prevents this.

We have imaged the ellipsoid body neurons for 150s and calculated motion and intensity fluctuation. New data is shown in Figure 3—figure supplement 4 and Video 4. We do not see major effects of motion on the fluorescence intensity. The reviewer is correct that when imaging for long periods of time z-drift is an issue. We have added a sentence to the text to mention this issue during chronic imaging.

3 – It is wonderful that the authors can image for hours using this technique, and critical to the breakthrough to know the extent to which smashing the head causes brain damage. We suggest repeating the elegant heat shock protein analysis (Figure 4 supplement 1) on flies chronically imaged by 2p and 3p.

We thank the reviewer for their kind words. We have repeated the HSP70 protein expression analysis for flies that were exposed to same amount of laser power as in chronic imaging experiments. Briefly, we exposed head-fixed flies to either 920nm or 1320nm laser wavelength for 6 minutes continuously under an objective lens. After the laser exposure flies were rested for 6 minutes then reimaged for another 6 minutes. We repeated the same procedure for 4 times, exposing the flies to in total 24 minutes of laser excitation as in chronically imaged flies. The control flies in these experiments were head-fixed but were not exposed to laser scanning under an objective lens. Our data suggests that there is no HSP70 protein expression when we use laser power 15mW, which is the upper limit of the laser power we used in all imaging experiments in this paper. These results are added to Figure 4—figure supplement 1 and mentioned in the text.

We also suggest testing whether flies removed from the imaging stage/preparation can mate and produce offspring, as this would be a simple demonstration that they are still capable of sensory perception, decision-making, and complex motor programs.

Thank you for this suggestion. To investigate whether head compression causes behavior deficits in flies, we quantified the copulation success of male flies that were previously head compressed. We found that head compression does not impact male courtship behavior and head-compressed male flies are still able to court and mate with females. These results are added to Figure 2—figure supplement 1A and mentioned in the text.

In all the points above, we don't require a particular result for publication. Rather, because this is a methods-focused paper, we need a fuller picture of both the possibilities and limits of this technical approach--what it can do, and what it still cannot do.

We understand the reviewer’s points. We hope that we have provided valuable information during the revisions which demonstrate the advantages and the limits of through-cuticle 2P and 3P imaging.